# Distilling and Adapting: A Topology-Aware Framework for Zero-Shot Interaction Prediction in Multiplex Biological Networks

**Alana Deng**[1], **Sugitha Janarthanan**[2], **Yan Sun**[1], **Zihao Jing**[1], **Pingzhao Hu**[1,2,*]
[1]Department of Computer Science, Western University, London, ON, Canada
[2]Department of Biochemistry, Western University, London, ON, Canada

## Abstract

Multiplex Biological Networks (MBNs), which represent multiple interaction types between entities, are crucial for understanding complex biological systems. Yet, existing methods often inadequately model multiplexity, struggle to integrate structural and sequence information, and face difficulties in zero-shot prediction for unseen entities with no prior neighbourhood information. To address these limitations, we propose a novel framework for zero-shot interaction prediction in MBNs by leveraging context-aware representation learning and knowledge distillation. Our approach leverages domain-specific foundation models to generate enriched embeddings, introduces a topology-aware graph tokenizer to capture multiplexity and higher-order connectivity, and employs contrastive learning to align embeddings across modalities. A teacher–student distillation strategy further enables robust zero-shot generalization. Experimental results demonstrate that our framework outperforms state-of-the-art methods in interaction prediction for MBNs, providing a powerful tool for exploring various biological interactions and advancing personalized therapeutics.

## 1 Introduction

Exploring complex interactions within biological networks is vital for advancing personalized medicine and drug discovery. These interactions enable targeted therapies and the study of disease pathways, calling for predictive models that are both precise and reliable. Moving beyond traditional single-layer analyses, multiplex networks provide a powerful framework to capture the multi-dimensional nature of biological systems, with each layer representing a distinct interaction type, providing a holistic picture of biological connectivity across diverse contexts.

Current predictive models (Kipf & Welling, 2016; Hamilton et al., 2017; Zhu et al., 2018; Rao et al., 2022; Tao et al., 2023; Dang & Vu, 2024), often rely on single-layer network analyses. Their limitations can be summarized in three areas: **(1) Handling Multiplex Data:** Single-layer representations discard relational heterogeneity and semantic distinctions between interaction types, overlooking context-dependent reasoning where interactions may rely on both relation type and broader network context. **(2) Integrating Multimodal Information:** These models have limited capacity to combine biological or chemical sequence features with network topology, which is essential for capturing complex biological interactions. **(3) Zero-Shot Prediction:** They struggle to predict interactions for unseen entities without prior neighborhood data, restricting generalization to novel biochemical entities.

To address these shortcomings, we propose **CAZI-MBN (Context-Aware and Zero-shot Interaction prediction in Multiplex Biological Networks)**, a framework tailored for multiplex biological interaction prediction. Its four **key contributions** include: **(1) Multiplex-oriented representation learning**, capturing interaction-type-specific, context-aware patterns across layered networks. **(2) Unified graph–sequence learning**, leveraging domain-specific LLMs for biochemically informed representations. **(3) Strategic knowledge distillation**, transferring latent knowledge from observed

---

*Corresponding author. Departments of Computer Science and Biochemistry, Western University, 1400 Western Road, London, Ontario N6G 2V4, Canada. E-mail: phu49@uwo.ca (P.H.).

to unobserved regions to enable zero-shot prediction. **(4) A Mixture of Experts (MoE)**, adaptively modeling diverse interaction types and label dependencies for multi-label prediction.

## 2   RELATED WORK

**Domain-Specific LLMs.**   Pre-trained language models have become useful tools for biological sequence analysis by learning patterns from large corpora of protein, molecular, or genomic sequences. ProtTrans (Elnaggar et al., 2021) and ESM-2 (Lin et al., 2023) enhance predictions of protein function and structure, while ChemBERTa (Chithrananda et al., 2020), trained on tens of millions of SMILES strings (Kim et al., 2023), captures chemical syntax and substructure patterns. DNABERT (Ji et al., 2021) and DNABERT-2 (Zhou et al., 2023) extend these approaches to genomic motifs and regulatory elements.

**Interaction Prediction and Graph Representation Learning.**   Interaction prediction in networks has advanced through graph representation learning, which embeds nodes or subgraphs into vector spaces preserving topology. Random-walk methods (Perozzi et al., 2014; Ribeiro et al., 2017; Dong et al., 2017; Gao et al., 2018; Grover & Leskovec, 2016) gave way to Graph Neural Networks (GNNs) (Scarselli et al., 2008; Wu et al., 2020; Zhang & Chen, 2018; Xia et al., 2024; Kim & Oh, 2022), including GCN (Kipf & Welling, 2016), Graph Transformer (Shi et al., 2020), and GraphSAGE (Hamilton et al., 2017) for neighborhood aggregation. Contrastive approaches like DGI (Veličković et al., 2018) further improve representations by maximizing mutual information. Relation-aware GNNs (Schlichtkrull et al., 2018; Chen et al., 2021) also expand these capabilities by parameterizing message passing with respect to edge types, enabling the model to learn relation-specific transformations and aggregate signals that vary across interaction modes.

**Multiplex Network Modeling.**   Multiplex networks can be modeled as Knowledge Graphs (KGs), which enable semantic reasoning (Walsh et al., 2020; Mohamed et al., 2021; Chen et al., 2023; Chandak et al., 2023) but flatten layered structures and lose interaction-type specificity, limiting zero-shot performance. Multi-layer network models (Park et al., 2020; Jing et al., 2021; Gallo et al., 2023; Gu et al., 2024) instead preserve intra- and inter-layer dependencies, supporting relation-aware learning and multiplex link prediction. Yet, in MBNs (Xiang et al., 2021; Yu et al., 2022; Valdeolivas et al., 2019), integrating sequence features and capturing complex biomedical dependencies across layers remains an open challenge.

**Knowledge Distillation.**   Knowledge distillation (Hinton, 2015) trains a smaller "student" to emulate a larger "teacher," enabling efficient deployment with minimal performance loss. It has proven effective in computer vision (Chen et al., 2017) and NLP (Sanh, 2019). However, its potential to MBNs and zero-shot generalization remains largely unexplored.

To summarize, existing approaches lack a systematic framework that unifies sequence embeddings with multiplex structures. Single-layer methods overlook relational heterogeneity, multiplex models underuse multimodal features, and most approaches rely on neighborhood data, with little effort devoted to exploring zero-shot generalization.

## 3   PROBLEM STATEMENT

**Definition:  Multiplex Networks.**   A multiplex network is a graph $\mathcal{G} = \{V, P, E_{intra}\} = \{\mathcal{G}_1, \ldots, \mathcal{G}_L\}$ where $V$ is a set of nodes across layers, $P$ is a set of entities, and $E_{intra}$ represents the set of intra-layer edges. Each layer $\mathcal{G}_i = \{P_i, V_i, E_i\}$ represents the graph of an interaction type in $\mathcal{G}$, where $P_i$, $V_i$, and $E_i$ are respectively the entity set, node set, and edge set for layer $i$, for $i = 1, \ldots, L$. In a multiplex network, an entity (e.g. gene, compound, protein) can be present as nodes in multiple layers, while node sets and the connectivity (edges) can differ from one layer to another. That is, $P_1, \ldots, P_L \in P$ are intersecting while $V_1, \ldots, V_L \in V$ and $E_1, \ldots, E_L \in E$ are disjoint. Each node $v \in V_i$ corresponds to an entity $p \in P$. This allows the representation of different types of relationships or interactions between the same set of entities. An inter-layer graph $G_{inter} = (V, E_{inter})$ can be formed by connecting the same entities in different layers of $\mathcal{G}$.

**Supra-Adjacency Matrix.** Given a multiplex network $\mathcal{G}$, there is a set of adjacency matrices $A = \{A_1, A_2, \dots, A_L\}$ where $A_i \in \mathcal{R}^{|V_i| \times |V_i|}$ is the layer-specific adjacency matrix for layer $i$. $A$ encodes the intra-layer links in $\mathcal{G}$. $G_{inter}$ can be represented by an inter-layer adjacency matrix $C \in \mathcal{R}^{|V| \times |V|}$. The supra-adjacency matrix, which encodes both intra-layer and inter-layer connections, is defined as

$$\hat{A} = \bigoplus_{i \in \{1, \dots, L\}} A_i + C, \tag{1}$$

where $\bigoplus$ denotes the direct sum. The illustration of an example multiplex network and its supra-adjacency matrix can be found in Figure 1.

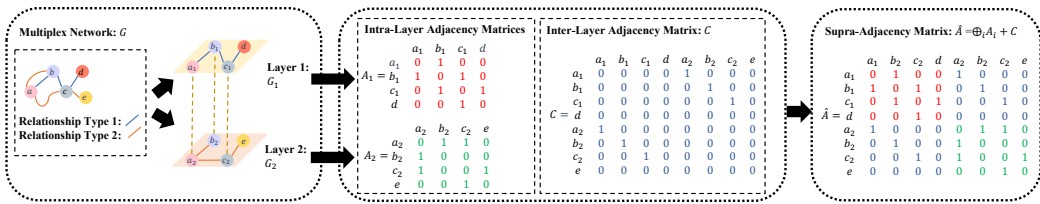

Figure 1: Illustration of an example multiplex network and its supra-adjacency matrix. $A_1$ and $A_2$ are the layer-specific adjacency matrices for layer 1 and 2, $C$ is the inter-layer adjacency matrix, and $\hat{A}$ is the supra-adjacency matrix that represents the example multiplex network $\mathcal{G}$.

**Generalizing Interaction Prediction in Multiplex Networks: From Seen to Unseen Entities.** Interaction prediction in multiplex networks is a multi-label classification task, since entity pairs can share multiple interaction types. Transductive settings predict missing links among seen entities, while zero-shot settings require predicting interactions for entirely unseen entities with no prior known neighborhood.

## 4 METHODS

The proposed CAZI-MBN framework addresses multiplex biological interaction prediction in both transductive and zero-shot settings. It integrates **domain-specific LLM embeddings** (see Supplementary Section A) with **topology-aware network representations**. A **Context-Aware Enhancement (CAE)** module (see Supplementary Section C) captures inter-layer dependencies, while a **MoE** model enables multi-label prediction and knowledge distillation supports zero-shot inference. The workflow is shown in Figure 2.

Training of the teacher model in CAZI-MBN is guided by a hybrid loss function:

$$\mathcal{L} = \mathcal{L}_{disc} + \mathcal{L}_{reg} + \mathcal{L}_{cls} + \beta \|\Theta\|^2, \tag{2}$$

where $\mathcal{L}_{disc}$ and $\mathcal{L}_{reg}$ are the discriminator's loss and the consensus regularizer's loss from the CAE, $\mathcal{L}_{cls}$ is a multi-label soft margin loss (see Supplementary Section B), and $\beta$ is the coefficient for $l2$ regularization applied to the trainable parameters $\Theta$. The student model is trained with the sum of a Mean Square Error (MSE) distillation loss $\mathcal{L}distill$ and $\mathcal{L}_{cls}$.

### 4.1 FEATURE REPRESENTATION

To represent the multimodal nature of MBN data, our approach integrates two complementary sources of information: sequence-based embeddings derived from **domain-specific pre-trained LLMs** and topology-based representations generated by a **Unified Graph Tokenizer (UGT)**.

#### 4.1.1 SEQUENCE EMBEDDING: DOMAIN-SPECIFIC LLMS

We use three pre-trained LLMs for sequence embeddings: ChemBERTa (Chithrananda et al., 2020) for drug/metabolite SMILES, DNABERT-2 (Zhou et al., 2023) for gene sequences, and ESM-2 (Lin et al., 2023) for proteins.

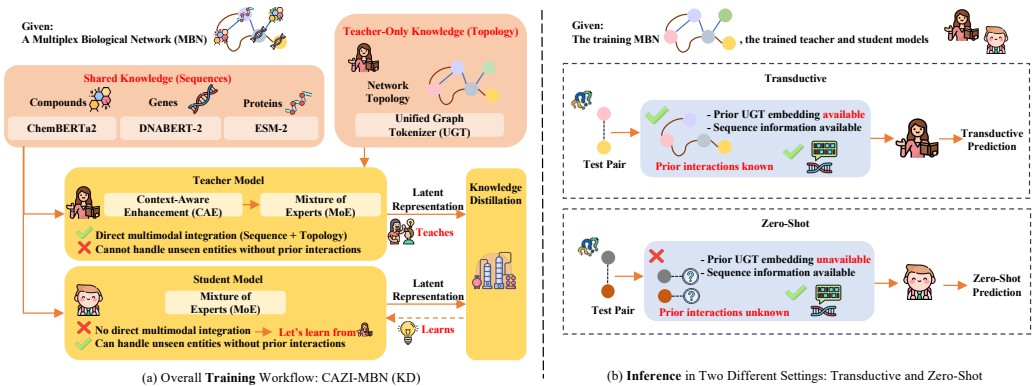

Figure 2: (a) The overall training workflow of CAZI-MBN. (b) Two inference settings in CAZI-MBN: Transductive and zero-shot.

### 4.1.2 UNIFIED GRAPH TOKENIZER (UGT)

The UGT generates generates topology, multiplexity, and high-order connectivity-aware node embedding from the supra-adjacency matrix $\hat{A}$ by (1) constructing a smoothed high-order matrix $\tilde{A} = \bar{A} + \bar{A}^2 + \cdots + \bar{A}^O$, with $\bar{A} = D^{-1/2}\hat{A}D^{-1/2}$, and (2) projecting topology-aware embeddings. For node $v \in V$:

$$e_v = \tilde{A}_{v,:}, \text{LN}(U\sqrt{\Sigma} \parallel V\sqrt{\Sigma}), \quad e_v \in \mathbb{R}^{d_u}, \tag{3}$$

where $U, V \in \mathbb{R}^{|V| \times d_u}$ and $\Sigma \in \mathbb{R}^{d_u \times d_u}$ are from the SVD of $\tilde{A}$.

### 4.2 CONTEXT-AWARE ENHANCEMENT (CAE)

The CAE module refines multiplex embeddings through node- and layer-level inter-layer attention combined with contrastive learning (Figure 3 and Supplementary Section C.). Each layer is encoded with a Graph Transformer, and inter-layer attention enables nodes to adaptively attend to counterparts across interaction types. A contrastive learning framework aligns embeddings by training a discriminator to distinguish real from perturbed edges, while a consensus regularizer learns a shared embedding that maximizes agreement across true layers and minimizes alignment with negative views. This design enhances generalization in complex biological systems by improving representation consistency and contextual relevance across the multiplex structure. A CAE forward pass is outlined in Algorithm 1.

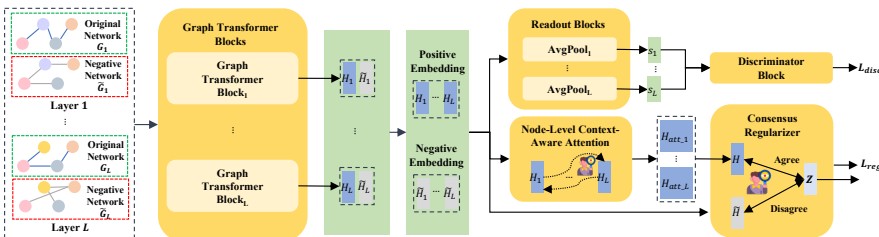

Figure 3: Illustration of the CAE module.

### 4.2.1 NODE-LEVEL CONTEXT-AWARE INTER-LAYER ATTENTION

Node-level context-aware inter-layer attention enables CAZI-MBN to adaptively weight an entity's representations across layers. The weight $a_n^{(p \leftarrow q)}$ quantifies how much node $n$ in layer $p$ attends to its counterpart in layer $q$.

---

**Algorithm 1** CAE Forward Pass

---

**Input:** Feature matrices $X_i$, graphs $G_i$, size $L$
**for** $i = 1$ **to** $L$ **do**
  $\tilde{G}_i \leftarrow NegSampling(G_i)$ {Negative sampling}
  $H_i \leftarrow GraphTransformer(X_i, G_i)$ {Feature matrix $X_i$}
  $H_{att\_i} \leftarrow NodeContextAwareAttention(H_i)$ {Node-level context-aware inter-layer attention}
  $\tilde{H}_i \leftarrow GraphTransformer(X_i, \tilde{G}_i)$
  $H_{edge_i} \leftarrow GetEdgeFeature(H_{att\_i}, G_i)$ {Edge embedding}
  $\tilde{H}_{edge_i} \leftarrow GetEdgeFeature(\tilde{H}_i, \tilde{G}_i)$ {Negative edge embedding}
  $s_i \leftarrow \sigma(AvgPool(H_{edge_i}))$
  $\text{logit}_{disc_i} \leftarrow Discriminator(H_{edge_i}, \tilde{H}_{edge_i}, s_i)$
  $\mathcal{L}_{disc_i} \leftarrow BCE(Y_{disc_i}, \text{logit}_{disc_i})$
**end for**
$H \leftarrow ATTENTION(H_{att\_1}, \ldots, H_{att\_L})$
$\tilde{H} \leftarrow ATTENTION(\tilde{H}_1, \ldots, \tilde{H}_L)$ {Layer-level attention aggregation}
$\mathcal{L}_{disc} \leftarrow SUM(\mathcal{L}_{disc_1}, \ldots, \mathcal{L}_{disc_L})$ {Discriminator's loss}
$Z \leftarrow ConsensusRegularizer(H, \tilde{H})$ {Updated embeddings}
$\mathcal{L}_{reg} \leftarrow 1 + CosineSim(H, Z) - CosineSim(\tilde{H}, Z)$ {Regularizer's loss}

---

$$a_n^{(p \leftarrow q)} = \text{softmax}\left(\frac{\sigma\left(\theta^{(p)} \cdot \left(H_n^{(p)} \otimes H_n^{(q)}\right)\right)}{\sum_{l=1, l \neq p}^{|L|} \sigma\left(\theta^{(p)} \cdot \left(H_n^{(p)} \otimes H_n^{(l)}\right)\right)}\right), \quad (4)$$

where $\theta^{(p)}$ is a learnable weight vector, and $\sigma(\cdot)$ is a sigmoid activation function.

### 4.2.2 LAYER-LEVEL CONTEXT-AWARE CONSENSUS EMBEDDING

To capture the varying importance of interaction types, CAZI-MBN applies an attention module that aggregates layer-specific node features into $H$ and its negative counterpart $\tilde{H}$. A context-aware contrastive learning-based consensus regularizer then learns a consensus embedding $Z \in \mathbb{R}^{|P| \times d}$ by maximizing agreement with $H$ and minimizing disagreement with $\tilde{H}$, using a cosine-similarity loss.

### 4.3 MIXTURE OF EXPERTS (MOE)

In MBNs, interaction prediction is a multi-label classification task since entity pairs can share multiple interaction types (Figure 2). To handle label sparsity and interdependence, we use a MoE framework where each expert captures distinct interaction patterns, and a gating network assigns weights based on the input. For embedding $\mathbf{h}$, experts $f_k(\mathbf{h})k = 1^K$ produce logits, with weights $\boldsymbol{a} = \text{softmax}(W_g\mathbf{h} + b_g)$; the final prediction is $\hat{\mathbf{Y}} = \sum k = 1^K \boldsymbol{a}_k f_k(\mathbf{h})$.

### 4.4 KNOWLEDGE DISTILLATION

We apply knowledge distillation in CAZI-MBN to enable zero-shot interaction prediction (Figure 2). The teacher leverages sequence and topology embeddings but depends on neighborhood context, while the student is topology-agnostic and relies only on sequence data. By minimizing the MSE between their latent representations, the student approximates the teacher's richer features, enabling effective zero-shot inference.

## 5 EXPERIMENTS

### 5.1 EXPERIMENT SETUP

#### 5.1.1 DATASETS

To our knowledge, no standardized benchmark datasets currently exist for evaluating models in the context of MBNs. To address this gap and provide reliable, well-validated benchmarks, we curated five MBNs from high-quality data sources (Cannon et al., 2024; Zdrazil et al., 2024; Li et al., 2024; Peng et al., 2025; Han et al., 2018) published in high-profile journals. A summary of the MBNs is provided in Table 1, with details on data collection and preprocessing available in Supplementary Section D.

Table 1: Dataset summary (ITs: Interaction Types)

|  | Context | # of Nodes | ITs | # of ITs |
|---|---|---|---|---|
| DGIdb | Drug-Gene | 1846 | Interaction mechanisms | 5 |
| ChEMBL | Compound-Bacteria | 9368 | Response phenotypes | 3 |
| PINNACLE | Protein-Protein | 7044 | Interaction in different cell types | 12 |
| MetaConserve | Protein-Metabolite | 329 | Conservation in bacteria strains | 4 |
| TRRUST | Gene-Gene | 2,862 | Gene regulartory roles | 3 |

#### 5.1.2 BASELINES AND EVALUATION

We benchmark CAZI-MBN against 11 models across four categories: **sequence-based** (Chen & Guestrin, 2016), **single-graph** (Kipf & Welling, 2016; Hamilton et al., 2017; Shi et al., 2020; Veličković et al., 2018), **multiplex graph-based/relation-aware** (Boccaletti et al., 2014; Park et al., 2020; Jing et al., 2021; Schlichtkrull et al., 2018; Chen et al., 2021), and **domain-specific** (Rao et al., 2022; Tao et al., 2023; Chen et al., 2019; Dang & Vu, 2024; Bai et al., 2023; Huang et al., 2021; Cui et al., 2022; Fakhry et al., 2023) (Table 2). Since most are not designed for multiplex networks, we train a separate classifier per interaction type. For zero-shot comparison, we also include knowledge-distilled variants of models lacking native zero-shot capability.

Table 2: Benchmark models

| Type | Benchmark Models |
|---|---|
| Sequence-based | MultiLayer Perceptron (MLP), XGBoost (XGB) |
| Single graph-based | GCN , GraphSAGE, Graph Transformer, DGI |
| Multiplex graph-based/Relation-Aware | MultiplexSAGE, DMGI, HDMI, R-GCN, R-GAT |
| Domain-Specific | (1) Drug-Gene & Compoud-Bacteria: CoSMIG, DGCL
(2) Protein-Protein: PIPR, xCAPT5
(3) Protein-Metabolite: DrugBAN, MolTrans
(4) Gene-Gene: DL-GGI, GENER |

We evaluate performance with Area Under the Receiver Operating Characteristic Curve (AUROC), Area Under the Precision-Recall Curve (AUPRC), and multi-label metrics including Hamming Score (HS) and Subset Accuracy (SA). Each experiment is conducted over three repeated trials, and we report the mean and Standard Deviation (SD) of each metric. For details of the evaluation metrics and experimental settings, see Supplementary Section E.1 and Supplementary Section E.2

### 5.2 EMPIRICAL ANALYSIS

We conduct a comprehensive evaluation of CAZI-MBN on five benchmark multiplex biological networks under both **transductive** (T) and **zero-shot** (ZS) settings. These two evaluation modes capture different forms of generalization: the transductive setting evaluates how well the model interpolates within a partially observed network, while the zero-shot setting tests its ability to infer interactions for nodes that are not seen during training.

Across the full set of 13 baselines, Tables 3, 4, 5, 6, and 7 present the strongest single-graph, multiplex, and domain-specific models for each dataset, with comparisons based on AUROC. These tables show circumstances where domain-oriented approaches are competitive and situations where multiplex methods benefit from access to multiple interaction types.

Detailed results for all benchmark models, including AUROC, AUPRC, HS, and SA in both evaluation settings, are included in Supplementary Section E.3.1. A focused analysis of performance on minority interaction types, which typically have fewer observations and more irregular training signals, is provided in Supplementary Section E.3.2. This analysis helps assess the stability of the models in label-imbalanced conditions that commonly arise in biological networks.

Figure 4 reports the average performance drop when each module of CAZI-MBN (LLMs, UGT, CAE, MoE) is removed. This summary highlights the contribution of each component in both transductive and zero-shot settings. Expanded ablation results and dataset-level discussions appear in Supplementary Section E.3.3. Figure 6 further reports module-wise performance drops for each dataset, and the error bars indicate variability across interaction types.

Additionally, we include a case study on IBD-related genes in Supplementary Section E.3.4, which illustrates how CAZI-MBN prioritizes biologically meaningful interactions in a real disease context. The parameter counts of the teacher and student models are presented in Supplementary Section E.3.5 to document the computational footprint of each component and to clarify the efficiency gains of the student architecture. To provide insight into how different interaction types contribute to the multiplex representation, the layer-specific attention weights from CAE are reported in Supplementary Section E.3.7.

Table 3: Evaluation of transductive and zero-shot multiplex interaction prediction on DGIdb (top model per category).

| Setting | Model | AUROC | AUPRC | HS | SA |
|---|---|---|---|---|---|
| T | Graph Transformer | 0.505±0.007 | 0.514±0.005 | 0.493±0.004 | 0.508±0.003 |
| | HDMI | 0.551±0.006 | 0.557±0.007 | 0.540±0.012 | 0.511±0.006 |
| | DGCL | 0.519±0.005 | 0.531±0.004 | 0.527±0.007 | 0.512±0.006 |
| | **CAZI-MBN** | **0.715±0.007** | **0.729±0.009** | **0.687±0.011** | **0.684±0.015** |
| ZS | Graph Transformer | 0.498±0.011 | 0.502±0.004 | 0.486±0.004 | 0.504±0.007 |
| | DMGI | 0.524±0.008 | 0.528±0.016 | 0.529±0.004 | 0.502±0.006 |
| | DGCL | 0.513±0.003 | 0.515±0.007 | 0.509±0.011 | 0.502±0.004 |
| | **CAZI-MBN** | **0.671±0.008** | **0.709±0.011** | **0.688±0.009** | **0.663±0.014** |

*See Table 10 in Supplementary Section E.3.1 for full results of all evaluated models.

Table 4: Evaluation of transductive and zero-shot multiplex interaction prediction on ChEMBL (top model per category).

| Setting | Model | AUROC | AUPRC | HS | SA |
|---|---|---|---|---|---|
| T | DGI | 0.651±0.019 | 0.719±0.005 | 0.738±0.012 | 0.673±0.011 |
| | HDMI | 0.663±0.012 | 0.762±0.014 | 0.789±0.012 | 0.730±0.006 |
| | CoSMIG | 0.661±0.021 | 0.732±0.011 | 0.755±0.009 | 0.702±0.025 |
| | **CAZI-MBN** | **0.812±0.008** | **0.863±0.006** | **0.889±0.014** | **0.757±0.011** |
| ZS | Graph Transformer | 0.628±0.011 | 0.720±0.008 | 0.725±0.013 | 0.628±0.014 |
| | DMGI | 0.652±0.021 | 0.745±0.015 | 0.756±0.007 | 0.711±0.009 |
| | CoSMIG | 0.641±0.011 | 0.733±0.007 | 0.729±0.010 | 0.683±0.005 |
| | **CAZI-MBN** | **0.791±0.018** | **0.839±0.015** | **0.857±0.011** | **0.723±0.009** |

*See Table 11 in Supplementary Section E.3.1 for full results of all evaluated models.

We note the following key observations: (1) **CAZI-MBN consistently outperforms all benchmarks in AUROC and AUPRC**, with improvements of 3.1–20.4% across the five multiplex networks in both transductive and zero-shot settings. These results indicate that the model is able to generalize

Table 5: Evaluation of transductive and zero-shot multiplex interaction prediction on PINNACLE (top model per category).

| Setting | Model | AUROC | AUPRC | HS | SA |
|---|---|---|---|---|---|
| T | Graph Transformer | 0.701±0.007 | 0.704±0.015 | 0.700±0.005 | 0.726±0.024 |
| | HDMI | 0.773±0.022 | 0.798±0.023 | 0.776±0.017 | 0.766±0.016 |
| | xCAPT5 | 0.781±0.011 | 0.804±0.014 | 0.726±0.012 | 0.752±0.015 |
| | **CAZI-MBN** | **0.831±0.018** | **0.845±0.011** | **0.751±0.009** | **0.772±0.013** |
| ZS | Graph Transformer | 0.687±0.019 | 0.690±0.017 | 0.680±0.018 | 0.705±0.019 |
| | HDMI | 0.748±0.025 | 0.776±0.021 | 0.757±0.024 | 0.742±0.014 |
| | xCAPT5 | 0.785±0.012 | 0.791±0.015 | 0.733±0.013 | 0.739±0.016 |
| | **CAZI-MBN** | **0.812±0.013** | **0.820±0.008** | **0.748±0.010** | **0.763±0.011** |

*See Table 12 in Supplementary Section E.3.1 for full results of all evaluated models.

Table 6: Evaluation of transductive and zero-shot multiplex interaction prediction on MetaConserve (top model per category).

| Setting | Model | AUROC | AUPRC | HS | SA |
|---|---|---|---|---|---|
| T | DGI | 0.668±0.020 | 0.673±0.019 | 0.665±0.010 | 0.669±0.010 |
| | HDMI | 0.726±0.015 | 0.729±0.021 | 0.715±0.006 | 0.718±0.015 |
| | DrugBAN | 0.737±0.010 | 0.714±0.012 | 0.710±0.019 | 0.711±0.008 |
| | **CAZI-MBN** | **0.752±0.020** | **0.744±0.012** | **0.779±0.008** | **0.656±0.014** |
| ZS | Graph Transformer | 0.651±0.007 | 0.660±0.020 | 0.652±0.015 | 0.648±0.015 |
| | HDMI | 0.709±0.017 | 0.710±0.014 | 0.704±0.014 | 0.707±0.009 |
| | DrugBAN | 0.692±0.017 | 0.705±0.020 | 0.707±0.015 | 0.708±0.011 |
| | **CAZI-MBN** | **0.738±0.015** | **0.722±0.010** | **0.749±0.020** | **0.653±0.009** |

*See Table 13 in Supplementary Section E.3.1 for full results of all evaluated models.

Table 7: Evaluation of transductive and zero-shot multiplex interaction prediction on TRRUST (top model per category).

| Setting | Model | AUROC | AUPRC | HS | SA |
|---|---|---|---|---|---|
| T | DGI | 0.841±0.014 | 0.844±0.011 | 0.845±0.009 | 0.847±0.006 |
| | HDMI | 0.878±0.008 | 0.879±0.014 | 0.883±0.005 | 0.881±0.004 |
| | GENER | 0.867±0.012 | 0.868±0.006 | 0.865±0.010 | 0.869±0.006 |
| | **CAZI-MBN** | **0.905±0.013** | **0.872±0.008** | **0.791±0.023** | **0.784±0.015** |
| ZS | DGI | 0.832±0.011 | 0.835±0.014 | 0.836±0.007 | 0.838±0.006 |
| | HDMI | 0.868±0.006 | 0.869±0.010 | 0.872±0.006 | 0.870±0.004 |
| | GENER | 0.857±0.007 | 0.858±0.008 | 0.855±0.006 | 0.859±0.007 |
| | **CAZI-MBN** | **0.899±0.017** | **0.869±0.013** | **0.775±0.019** | **0.764±0.007** |

*See Table 14 in Supplementary Section E.3.1 for full results of all evaluated models.

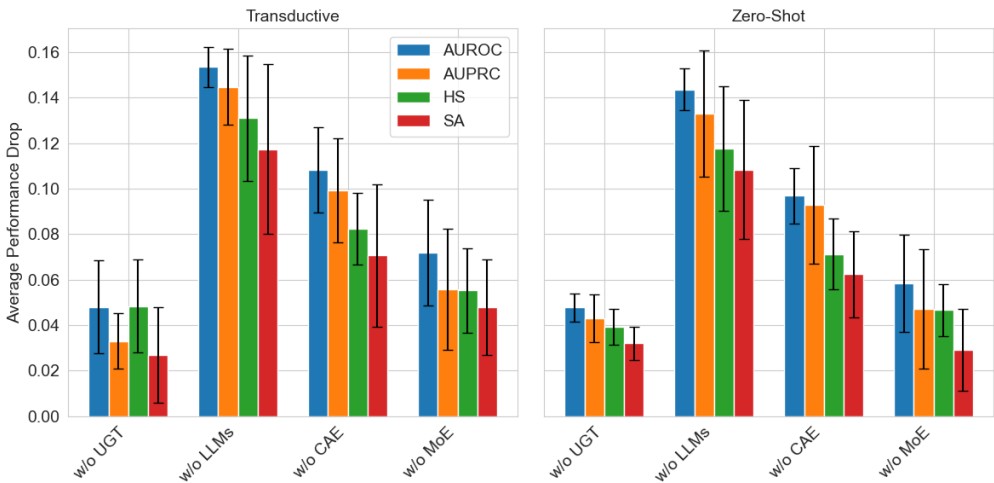

Figure 4: Average metric-wise performance drop across all five datasets when each module is ablated.

across multiple biological contexts. (2) **Flexible calibration strategies can help close HS and SA gaps.** For datasets where CAZI-MBN lags on HS/SA, our post-hoc per-label calibration experiment (Supplementary Section E.3.6) shows that performance improves noticeably with more flexible thresholds, suggesting the benefits of a per-label calibration strategy, as interaction types often have different score distributions. (3) **Multiplex and domain-specific models** such as DMGI, HDMI, CoSMIG, xCAPT5, and GENER usually outperform single-graph baselines, underscoring the importance of incorporating relation-specific structure and biological context. (4) The **interaction type-specific analysis** (Supplementary Section E.3.2) shows that CAZI-MBN performs well on minority interaction types in zero-shot settings, suggesting that the model remains stable even when training data for certain interaction types are scarce. (5) **Ablation studies** (Supplementary Section E.3.3 and Figure 4) show that all components make contributions to performance. LLMs contribute the largest gains (15–20% AUROC), followed by CAE (7–10%) and UGT/MoE (5–8%). These results suggest that the overall design relies on several complementary sources of information. (6) The **case study** (Table 18 in Supplementary Section E.3.4) shows that CAZI-MBN recovers a substantial proportion of known interactions (82.7% DGIdb, 85.7% TRRUST, 75.1% PINNACLE, 62.5% MetaConserve) and identifies literature-supported IBD-related links such as GAPDH–Omigapil (Foley et al., 2024; Zhou et al., 2015; Özsoy et al., 2022). These results demonstrates the model's biologically coherent predictions in a zero-shot setting

## 6 CONCLUSION, LIMITATIONS, AND FUTURE WORK

We present CAZI-MBN, a zero-shot framework for interaction prediction in multiplex biological networks that combines domain-specific LLMs, multiplex-aware embeddings, contrastive attention, and teacher-student knowledge distillation, achieving strong performance and robust generalization to unseen entities. To our knowledge, this is the first framework specifically designed for zero-shot multiplex interaction prediction. We also curate five high-quality MBN datasets to address the lack of standardized benchmarks. CAZI-MBN shows promise for accelerating drug discovery and antibiotic research by enabling interaction prediction for novel biochemical entities, and its flexible design is applicable to other biomedical domains.

Nonetheless, the framework has limitations. In particular, it does not yet incorporate structural data (e.g., genomic, protein, or compound 3D information), which constrains fine-grained biological modeling. Future work will integrate such structural information and extend the framework to more complex settings such as cross-species networks.

## ETHICS STATEMENT

This work introduces a framework for predicting interactions in multiplex biological networks, with potential to accelerate discovery in areas such as drug development and precision medicine. We identify no direct ethical risks and are committed to responsible dissemination, supporting the ethical use of biological data and AI technologies.

## REPRODUCIBILITY STATEMENT

The source code for CAZI-MBN is publicly available at `https://github.com/alanadeng/CAZI-MBN`. Details of dataset preprocessing are provided in Supplementary Section C, while the experimental protocol and evaluation setup are described in Supplementary Section E.1 and Supplementary Section E.2.

## ACKNOWLEDGEMENT

This work was supported in part by the Canada Research Chairs Tier II Program (CRC-2021-00482), the Canadian Institutes of Health Research (PLL 185683, PJT 190272, PJT204042, CFA - 205059), the Natural Sciences, Engineering Research Council of Canada (RGPIN-2021-04072,ALLRP 602759-24) and The Canada Foundation for Innovation (CFI) John R. Evans Leaders Fund (JELF) program (#43481).

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

# DISTILLING AND ADAPTING: A TOPOLOGY-AWARE FRAMEWORK FOR ZERO-SHOT INTERACTION PREDICTION IN MULTIPLEX BIOLOGICAL NETWORKS

## A  DOMAIN-SPECIFIC PRE-TRAINED LLMS

**DNABERT-2 (Zhou et al., 2023).**   DNABERT-2 is a domain-adapted language model designed for nucleotide sequences, extending the capabilities of the original DNABERT by incorporating a deeper architecture and pretraining on a larger corpus of genomic data. It tokenizes DNA sequences using k-mers and leverages the transformer architecture to capture regulatory patterns and sequence dependencies across long genomic regions. DNABERT-2 has demonstrated improved performance on a range of genomics tasks such as promoter recognition, splice site prediction, and enhancer identification. In our work, we use the public checkpoint `zhihan1996/DNABERT-2-117M` (available at `https://huggingface.co/zhihan1996/DNABERT-2-117M`).

**ChemBERTa (Chithrananda et al., 2020).**   ChemBERTa is a transformer-based language model pre-trained on SMILES strings representing chemical structures. By treating SMILES as a specialized language, ChemBERTa facilitates downstream tasks such as molecular property prediction, reaction classification, and drug-likeness assessment. Its embeddings serve as chemically informed representations that integrate both syntactic and structural aspects of small molecules.

In our work, we use the public checkpoint `DeepChem/ChemBERTa-77M-MLM` (available at `https://huggingface.co/DeepChem/ChemBERTa-77M-MLM`).

**ESM-2 (Lin et al., 2023).** ESM-2 is a protein language model developed by Meta AI, pre-trained on hundreds of millions of protein sequences using a masked language modeling objective. It models the grammar of amino acid sequences and captures biologically relevant features such as structural motifs, binding sites, and evolutionary conservation without explicit supervision. Compared to its predecessor, ESM-2 offers deeper transformer layers and larger-scale training, enabling superior performance across protein structure prediction, variant effect analysis, and protein-protein interaction tasks. Its embeddings encode rich evolutionary and functional signals useful for downstream applications in proteomics and systems biology. In our work, we use the public checkpoint `facebook/esm2_t30_150M_UR50D` (available at `https://huggingface.co/facebook/esm2_t30_150M_UR50D`).

## B  MULTI-LABEL SOFT MARGIN LOSS (MLSML)

The MLSML is defined as:

$$\mathcal{L}_{cls} = -\frac{1}{N} \sum_{i=1}^{N} \sum_{j=1}^{N_C} y_{ij} \cdot \log(\sigma(x_{ij})) + (1 - y_{ij}) \cdot \log(1 - \sigma(x_{ij})), \tag{5}$$

where $N$ is the number of samples, $N_C$ is the number of classes, $y_{ij}$ is a binary indicator for the presence of class $j$ in sample $i$, $x_{ij}$ is the raw model output for class $j$ of sample $i$, and $\sigma$ is the sigmoid function.

## C  THE CAE MODULE

For each multiplex layer $G_i$, a Graph Transformer encodes node features $X_i$ into embeddings $H_i$. Negative sampling generates perturbed graphs $\tilde{G}_i$ and corresponding embeddings $\tilde{H}_i$. Context-aware node-level inter-layer attention refines $H_i$ into $H_{\text{att}_i}$, from which edge features are extracted. A discriminator differentiates real edges from corrupted ones. Embeddings are then aggregated across layers with attention, and a consensus regularizer produces updated embeddings $Z$.

The CAE module starts with layer-specific node embedding. For each multiplex layer $G_i(i \in \{1, \ldots, L\})$ in the multiplex network, a Graph Transformer module first updates the layer-specific node embeddings according to:

$$H_i^{(l+1)} = \sigma \left( \text{LN} \left( \text{MHA} \left( H_i^{(l)} \right) + H_i^{(l)} \right) \cdot W_i^{(l)} \right), \tag{6}$$

where LN represents layer normalization, $H_i^{(l)} \in \mathbb{R}^{|V_i| \times d}$ denotes the feature matrix of multiplex layer $G_i$ at Graph Transformer layer $l$, $W_i^{(l)}$ is the layer-specific weight matrix, MHA performs multi-head self-attention over the node features, and $d$ represents the output dimension. To generate the layer-specific negative node embedding $\tilde{H}_i$, negative sampling is applied to $G_i$, and the same Graph Transformer module as defined in Equation 6 processes the negative network $\tilde{G}_i$.

The edge feature matrix $H_{edge_i} \in \mathbb{R}^{|E_i| \times 2d}$ and the negative edge feature matrix $\tilde{H}_{edge_i} \in \mathbb{R}^{|E_i| \times 2d}$ are constructed by concatenating the node features of adjacent nodes within $G_i$ and $\tilde{G}_i$, respectively. A high-level graph summary vector $s_i$ is computed to encapsulate the layer-level edge information for each $G_i$ using average pooling:

$$s_i = \sigma(\text{AvgPool}(H_{\text{edge}_i})), \quad (s_i \in \mathbb{R}^{2d}) \tag{7}$$

where $\sigma$ is the logistic sigmoid nonlinearity.

The layer-specific BCE loss $\mathcal{L}_{\text{disc}_i}$ can be calculated using the edge embedding matrix $H_{\text{edge}_i}$, its summary representation $s_i$, and its corresponding negative matrix $\tilde{H}_{\text{edge}_i}$:

$$\mathcal{L}_{\text{disc}_i} = \sum_{e_j \in E_i} \log \mathcal{D}(H_{\text{edge}_i, j}, s_i) + \sum_{e_q \in \hat{E}_i} \log(1 - \mathcal{D}(\tilde{H}_{\text{edge}_i, q}, s_i)) \tag{8}$$

Table 8: Summary Statistics of Datasets.

| Dataset | Interaction Types | # Nodes | # Edges | # Interaction Types | Multiplex Rate (%) |
|---|---|---|---|---|---|
| DGIdb | Inhibitor (7,004) Modulator (1,439) Agonist (915) Activator (104) Other (170) | 1,846 | 8,443 | 5 | 1.34 |
| ChEMBL | S (120,265) I (33,925) R (94,392) | 9,368 | 248,582 | 3 | 14.45 |
| PINNACLE | Enterocyte-LI (4,089) Enterocyte-SI (7,433) Goblet-LI (3,073) Goblet-SI (6,258) Paneth-LI (8,291) Paneth-SI (15,093) CD4-Helper (6,893) CD4-Memory (7,190) CD8-Cytotoxic (8,172) Treg (15,626) B (9,261) Memory-B (10,064) | 7,044 | 101,443 | 12 | 23.26 |
| MetaConserve | Campylobacterales (109) Desulfovibrionales (98) Fusobacteriales (94) Veillonellales (76) | 329 | 377 | 4 | 27.10 |
| TRRUST | Activation (3,149) Repression (1,922) Unknown (4,325) | 2,862 | 9,396 | 3 | 8.9 |

$$\text{Multiplex Rate} = \frac{\#\text{ node pairs that appear in } \geq 2 \text{ layers}}{\#\text{ unique node pairs in the multiplex graph}} \times 100\%.$$

where $H_{\text{edge}_i,j}$ specifies the feature vector for edge $e_j$, $\hat{E}_i$ represents the edge set of $\tilde{G}_i$, and $\mathcal{D}$ is a discriminator function that evaluates patch-summary representation pairs. The discriminator $\mathcal{D}$ is defined as:

$$\mathcal{D}(H_{\text{edge}_i,j}, s_i) = \sigma(H_{\text{edge}_i,j} M_i s_i) \qquad (9)$$

where $\sigma$ signifies the logistic sigmoid function, and $M_i \in \mathbb{R}^{2d \times 2d}$ is a trainable scoring matrix.

## D  DATASETS AND PREPROCESSING

Table 8 shows the summary statistics of five MBNs.

### D.1  DGIDB

The Drug–Gene Interaction Database (DGIdb) (Cannon et al., 2024) aggregates comprehensive information on drug–gene interactions and gene druggability from diverse curated sources.

The multiplex nature of DGIdb reflects the biological complexity wherein a single drug–gene pair may exhibit multiple modes of interaction. For example, a drug may function as an agonist under certain conditions, but act as an inhibitor, modulator, or activator in others. These variations can arise due to factors such as dosage, cellular context, or the presence of cofactors.

During preprocessing, entries lacking interaction type annotations were removed, and semantically similar interaction types were merged to reduce redundancy.To emphasize higher-confidence associations, we excluded interactions with scores falling in the lowest 15% quantile. We remove

drugs that appear less than 7 times and genes that appear less than 4 times. Furthermore, to address class imbalance, the four least prevalent interaction categories were combined into a single class, simplifying downstream modeling and evaluation. We retained five broad mechanism categories after preprocessing: agonist, activator, modulator, inhibitor/blocker, and other.

## D.2 CHEMBL

ChEMBL (Zdrazil et al., 2024) is a manually curated database of bioactive molecules with drug-like properties. In ChEMBL, different Minimum Inhibitory Concentration (MIC) values for the same compound-bacteria pair indicate that the bacterium may exhibit varying levels of susceptibility to the compound under different experimental conditions or at different times. A compound bacteria pair with multiple MIC values might show different response phenotype - Susceptible (S), Intermediate (I), or Resistant (R), based on environmental or genetic factors. This variability can inform clinicians of potential shifts in bacterial resistance over time, the need for adjusted drug dosages, or the importance of considering strain-specific responses when choosing treatment strategies.

Our benchmark is a filtered, higher-quality subset of ChEMBL. We retained only assays with MIC-related measurements that could be converted into unified units, kept entries with valid canonical SMILES to ensure consistent compound representation, and applied a minimum-frequency filter by keeping only compounds and bacterial strains that appear more than 11 times. These steps reduce noise and sparsity, yielding a coherent dataset suitable for reliable multiplex interaction prediction rather than reflecting the full size of the ChEMBL database.

We preprocess ChEMBL into a multiplex network by (1) standardizing the standard units to "ug.mL-1"; (2) defining cutoffs for S, I, and R:

- Susceptible (S): MIC $\leq 4$;
- Intermediate (I): $4 < \text{MIC} \leq 16$;
- Resistant (R): MIC $< 16$.

## D.3 PINNACLE

We curated the Protein Network-based Algorithm for Contextual Learning (PINNACLE) dataset from (Li et al., 2024). The original dataset captures context-aware protein-protein interactions across 156 cell type contexts spanning 62 tissues, derived from the Tabula Sapiens single-cell transcriptomic atlas. Protein-protein interactions can differ across cell types because protein expression levels, post-translational modifications, subcellular localizations, and interacting partners vary depending on the cellular environment. A protein may be active in one cell type but absent or inactive in another, or it may participate in different pathways depending on the cell's functional role. These context-dependent variations are critical for understanding cell-specific mechanisms of health and disease.

For our study, we selected 12 cell types specifically associated with Inflammatory Bowel Disease (IBD) to construct a cell type-specific multiplex protein-protein interaction network. This enables us to investigate cell-type-dependent protein interactions relevant to IBD pathogenesis and therapeutic targeting, capturing the cellular heterogeneity of intestinal inflammation.

We selected the following cell types based on their established roles in IBD:

- **Enterocytes** (*Enterocyte of epithelium of large intestine* and *Enterocyte of epithelium of small intestine*): These epithelial cells form the intestinal barrier and regulate nutrient absorption and immune signaling, both of which are disrupted in IBD.

- **Goblet cells** (*Small intestine goblet cell* and *Large intestine goblet cell*): Goblet cells produce mucins that maintain the protective mucus layer; their depletion is linked to barrier dysfunction and increased inflammation in IBD.

- **Paneth cells** (*Paneth cell of epithelium of small intestine* and *Paneth cell of epithelium of large intestine*): These cells secrete antimicrobial peptides and are often dysregulated or dysfunctional in inflamed or chronically affected IBD tissue.

- **CD4$^+$ helper T cells** (*CD4-positive helper T cell*): Key mediators of adaptive immunity and cytokine signaling, CD4$^+$ T cells are known to drive pathogenic responses in IBD.

- **CD8$^+$ cytotoxic T cells** (*CD8-positive alpha-beta cytotoxic T cell*): These cells contribute to epithelial damage by directly killing intestinal epithelial cells in active IBD.

- **Regulatory T cells (Tregs)** (*Regulatory T cell*): Tregs suppress immune responses and maintain tolerance; impaired Treg function is frequently observed in IBD patients.

- **Memory T cells** (*CD4-positive alpha-beta memory T cell*): These cells retain antigen-specific memory and are implicated in the persistent immune activation characteristic of chronic IBD.

- **B cells** (*B cell*): B cells contribute to antigen presentation and cytokine production and can influence both protective and pathogenic pathways in IBD.

- **Memory B cells** (*Memory B cell*): These cells are involved in long-term humoral responses and are elevated in IBD-associated lymphoid aggregates and inflamed mucosa.

## D.4 METACONSERVE

We curated a cell type-specific protein–metabolite MBN dataset, which we named MetaConserve, from (Peng et al., 2025), which systematically mapped high-confidence ligand interactions in *Escherichia coli* using affinity purification mass spectrometry and structural modeling. This dataset provides a valuable foundation for exploring metabolite-mediated regulatory mechanisms at the host–microbe interface.

To investigate the evolutionary conservation of these protein–metabolite interactions in the context of IBD, we selected four representative bacterial orders: *Campylobacterales*, *Desulfovibrionales*, *Fusobacteriales*, and *Veillonellales*, all of which have been implicated in IBD pathogenesis. *Campylobacterales* (e.g., *Campylobacter concisus*) are frequently enriched in inflamed mucosa and are associated with epithelial barrier disruption and proinflammatory signaling. *Desulfovibrionales* are sulfate-reducing bacteria that produce hydrogen sulfide, a genotoxic metabolite elevated in IBD patients and implicated in mucosal injury. *Fusobacteriales*, particularly *Fusobacterium nucleatum*, are linked to mucosal inflammation and have been shown to exacerbate disease through immune modulation and epithelial adhesion. *Veillonellales* are enriched in dysbiotic gut microbiota and are associated with IBD disease activity and disrupted bile acid metabolism.

We leveraged conservation scores provided in the original dataset to identify protein–metabolite interactions that are preserved across these four orders, retaining only those with a conservation score $\geq 2$ to ensure high-confidence evolutionary retention. These conserved interactions define four distinct interaction types in our multiplex network, enabling a strain-aware investigation of microbial metabolic activity relevant to IBD onset and progression.

## D.5 TRRUST

The Transcriptional Regulatory Relationships Unraveled by Sentence-based Text mining (TRRUST) (Han et al., 2018) dataset is a curated database of human and mouse transcriptional regulatory interactions. It compiles experimentally validated regulatory relationships between transcription factors (TFs) and their target genes, extracted through literature mining and manual curation. Each entry in TRRUST specifies the regulator (TF), the target gene, the mode of regulation (activation or repression), and the supporting evidence. TRRUST is widely used in systems biology and regulatory network analysis, offering a high-confidence resource for studying gene regulatory mechanisms, constructing transcriptional networks, and evaluating computational predictions of TF-target interactions.

In our experiments, we use the human transcriptional regulatory data from TRRUST. As a gene can exert multiple regulatory effects on another gene, such as activation, repression, or an undefined role, and the resulting network is modeled as a multiplex graph. This structure captures the complexity of transcriptional regulation, where interactions may not be limited to a single regulatory type and can reflect more nuanced or context-dependent mechanisms.

# E  EXPERIMENTS AND EVALUATION

## E.1  EVALUATION METRICS

Let there be $N$ samples and $N_c$ possible labels. We denote the ground-truth labels for sample $i$ by a binary vector $\mathbf{y}_i = (y_{i,1}, y_{i,2}, \ldots, y_{i,N_C}) \in \{0,1\}^{N_C}$, and the predicted labels by $\hat{\mathbf{y}}_i = (\hat{y}_{i,1}, \hat{y}_{i,2}, \ldots, \hat{y}_{i,N_C})$. Below, we briefly define several common evaluation metrics.

**Area Under the ROC Curve (AUROC).**  For each label $j$, we can compute the Receiver Operating Characteristic (ROC) curve by plotting the True Positive Rate (TPR) versus the False Positive Rate (FPR) at various threshold settings. The area under this curve for label $j$ is denoted $\mathrm{AUROC}_j$. The AUROC in multi-label classification is calculated by macro-average:

$$\mathrm{AUROC} = \frac{1}{N_C} \sum_{j=1}^{N_C} \mathrm{AUROC}_j. \tag{10}$$

**Area Under the Precision-Recall Curve (AUPRC).**  For each label $j$, the Precision-Recall curve is constructed by plotting Precision versus Recall at different thresholds. The area under this curve for label $j$ is $\mathrm{AUPRC}_j$. The macro-average of AUPRC for multi-label classification is:

$$\mathrm{AUPRC} = \frac{1}{N_C} \sum_{j=1}^{N_C} \mathrm{AUPRC}_j. \tag{11}$$

**Hamming Score (HS).**  The HS measures the fraction of correctly predicted labels across all samples and all labels. Often, the Hamming Loss (HL) is defined first:

$$\mathrm{HL} = \frac{1}{N \cdot N_C} \sum_{i=1}^{N} \sum_{j=1}^{N_C} \mathbf{1}\big(y_{i,j} \neq \hat{y}_{i,j}\big), \tag{12}$$

where $\mathbf{1}(\cdot)$ is the indicator function, which is 1 if the condition is true and 0 otherwise. The Hamming Score is then

$$\mathrm{HS} = 1 - \mathrm{HL} = 1 - \frac{1}{N \cdot N_C} \sum_{i=1}^{N} \sum_{j=1}^{N_C} \mathbf{1}\big(y_{i,j} \neq \hat{y}_{i,j}\big). \tag{13}$$

**Subset Accuracy (SA).**  SA, also known as Exact Match Ratio, measures the fraction of samples for which all labels match exactly. Formally,

$$\mathrm{SA} = \frac{1}{N} \sum_{i=1}^{N} \mathbf{1}\big(\mathbf{y}_i = \hat{\mathbf{y}}_i\big). \tag{14}$$

Here, $\mathbf{1}\big(\mathbf{y}_i = \hat{\mathbf{y}}_i\big)$ is 1 if and only if every label is predicted correctly for sample $i$, and 0 otherwise.

## E.2  EXPERIMENTAL SETTINGS.

We divide our dataset into training, validation, and test sets with proportions of 75%, 15%, and 15%, respectively. Each experiment is repeated three times, and we report the mean and SD of each evaluation metric. In multiplex networks, negative sampling is defined over node pairs rather than individual layers. We treat a node pair $(u, v)$ as a positive instance if it appears in at least one layer, and as a negative instance if it does not appear in any layer, corresponding to an all-zero label vector. As in Schlichtkrull et al. (2018) and Kim & Oh (2022), we use a standard uniform random negative sampling strategy, where the ratio is 1:1, which means that each positive node pair is matched with one randomly selected non-interacting pair drawn from the set of all unobserved pairs.

To generate LLM embeddings for bacteria entities, their genomes are segmented into fixed-length windows (length=1024) and fed into DNABERT-2. We average the embedding outputs across windows to obtain a compact genome-level representation.

In the transductive setting, entities in the training, validation, and test sets overlap, meaning that all nodes are present during training but the edges are partitioned. As a result, models can exploit topological context from the training graph when making predictions, and the task is to infer unseen interactions among already observed entities. This setting reflects scenarios where the biological entities are known, but their interaction space is incomplete.

In contrast, the zero-shot setting enforces a strict disjoint entity-based split, where entities in the validation and test sets are entirely excluded from the training graph. These entities do not appear as endpoints or even as neighbors during training, ensuring that the model cannot rely on structural information about them. Instead, predictions must be made solely from their intrinsic features (e.g., sequence embeddings), without any prior connectivity. The task therefore involves predicting interactions among completely novel entities, simulating realistic discovery scenarios where new drugs, genes, or proteins enter the system with no known interactions. This strict separation guarantees that no information leakage occurs between training and test sets in either setting.

Hyperparameters are selected via grid search, and models are trained using the Adam optimizer. All experiments are conducted on a single NVIDIA A100 Tensor Core GPU with 80GB of RAM. The training deatils of CAZI-MBN are presented in Table 9.

For benchmark models using node attributes, compounds are encoded with Morgan fingerprints from SMILES, and genes/genomes/proteins with $k$-mer features ($k = 6$) from sequence data. To provide a more detailed comparison, we additionally include results from MLP and XGB benchmarks that use the same sequence LLM embeddings as our model.

For R-GCN and R-GAT, each multiplex layer is treated as a distinct relation type. R-GCN performs relation-specific message passing by applying a separate transformation to each relation type and aggregating the resulting messages into unified node embeddings. R-GAT also computes relation-specific attention coefficients on these relation-labeled edges and combines the weighted messages to update node embeddings. In zero-shot setting, where traditional graph-based models cannot inherently predict interactions involving entities with no observed training interactions, we adopt knowledge-distilled variants of these models. Specifically, the original graph-based models serve as teacher models, while a two-layer MLP classifier chain acts as the student model. This setup facilitates performance comparisons in zero-shot learning.

Table 9: Training details of CAZI-MBN.

| Component | Value/Detail |
|---|---|
| Optimizer | Adam |
| Learning rate | 1e-4 |
| Batch size | 256 |
| Number of epoches | 10000 |
| Early stopping | Yes |
| Order of UGT | 5 |
| UGT output dimension | 128 |
| Readout (CAE) | SAGPool |
| Latent dimension (knowledge distillation) | 32 |

### E.3 SUPPLEMENTAL EXPERIMENTAL RESULTS

### E.3.1 EMPIRICAL ANALYSIS

Tables 10, 11, 12, 13, and 14 present the performance comparisons between CAZI-MBN and all benchmark models under both transductive ("T") and zero-shot ("ZS") settings across five MBNs. All metrics are reported in Mean $\pm$ SD over three repeated trails.

In terms of robustness of CAZI-MBN over multiplex networks with different sparsity levels, our analysis shows that the model consistently maintains strong zero-shot performance even when the structural density of the multiplex graph varies across datasets. There is naturally some impact: if the multiplex graph is denser, the teacher can extract richer structural signals, which can strengthen the distilled representations. Sparsity can limit the amount of structure the teacher learns, but it is

Table 10: Evaluation of transductive and zero-shot multiplex interaction prediction on DGIdb.

| Setting | Model | AUROC | AUPRC | HS | SA |
|---|---|---|---|---|---|
| T | XGB | 0.350±0.010 | 0.389±0.011 | 0.579±0.011 | 0.421±0.011 |
| | MLP | 0.284±0.013 | 0.375±0.005 | 0.609±0.009 | 0.407±0.011 |
| | XGB (LLM) | 0.517±0.012 | 0.572±0.009 | 0.541±0.014 | 0.488±0.010 |
| | MLP (LLM) | 0.511±0.011 | 0.561±0.008 | 0.536±0.010 | 0.503±0.012 |
| | GCN | 0.482±0.010 | 0.493±0.005 | 0.511±0.003 | 0.491±0.003 |
| | GraphSAGE | 0.447±0.005 | 0.467±0.003 | 0.527±0.006 | 0.473±0.006 |
| | Graph Transformer | 0.505±0.007 | 0.514±0.005 | 0.493±0.004 | 0.508±0.003 |
| | DGI | 0.503±0.002 | 0.507±0.004 | 0.510±0.003 | 0.500±0.003 |
| | MultiplexSAGE | 0.532±0.007 | 0.539±0.004 | 0.549±0.005 | 0.507±0.005 |
| | DMGI | 0.547±0.002 | 0.542±0.003 | 0.533±0.005 | 0.505±0.002 |
| | HDMI | 0.551±0.006 | 0.557±0.007 | 0.540±0.012 | 0.511±0.006 |
| | R-GCN | 0.536±0.018 | 0.541±0.009 | 0.530±0.014 | 0.508±0.011 |
| | R-GAT | 0.544±0.007 | 0.549±0.012 | 0.521±0.017 | 0.514±0.021 |
| | CoSMIG | 0.517±0.003 | 0.523±0.002 | 0.522±0.003 | 0.509±0.001 |
| | DGCL | 0.519±0.005 | 0.531±0.004 | 0.527±0.007 | 0.512±0.006 |
| | **CAZI-MBN** | **0.715±0.007** | **0.729±0.009** | **0.687±0.011** | **0.684±0.015** |
| ZS | XGB | 0.332±0.013 | 0.361±0.008 | 0.513±0.013 | 0.401±0.005 |
| | MLP | 0.258±0.012 | 0.307±0.007 | 0.590±0.011 | 0.347±0.005 |
| | XGB (LLM) | 0.487±0.011 | 0.522±0.010 | 0.517±0.015 | 0.496±0.006 |
| | MLP (LLM) | 0.494±0.014 | 0.525±0.006 | 0.522±0.012 | 0.492±0.007 |
| | GCN | 0.448±0.008 | 0.479±0.004 | 0.502±0.005 | 0.473±0.007 |
| | GraphSAGE | 0.419±0.006 | 0.434±0.004 | 0.518±0.011 | 0.464±0.005 |
| | Graph Transformer | 0.498±0.011 | 0.502±0.004 | 0.486±0.004 | 0.504±0.007 |
| | DGI | 0.489±0.004 | 0.504±0.011 | 0.502±0.003 | 0.477±0.006 |
| | MultiplexSAGE | 0.518±0.008 | 0.520±0.013 | 0.537±0.009 | 0.504±0.006 |
| | DMGI | 0.524±0.008 | 0.528±0.016 | 0.529±0.004 | 0.502±0.006 |
| | HDMI | 0.507±0.004 | 0.536±0.012 | 0.530±0.011 | 0.507±0.008 |
| | R-GCN | 0.510±0.015 | 0.514±0.008 | 0.525±0.028 | 0.500±0.013 |
| | R-GAT | 0.512±0.012 | 0.518±0.014 | 0.519±0.015 | 0.507±0.020 |
| | CoSMIG | 0.509±0.012 | 0.514±0.005 | 0.516±0.013 | 0.500±0.020 |
| | DGCL | 0.513±0.003 | 0.515±0.007 | 0.509±0.011 | 0.502±0.004 |
| | **CAZI-MBN** | **0.671±0.008** | **0.709±0.011** | **0.688±0.009** | **0.663±0.014** |

*Benchmarks are grouped into four categories: (1) Sequence-based (XGB, MLP);(2) Single graph-based (GCN, GraphSAGE, Graph Transformer, DGI); (3) Multiplex graph-based/Relation-aware (MultiplexSAGE, DMGI, HDMI, R-GCN, R-GAT); and (4) Domain-specific (others).

Table 11: Evaluation of transductive and zero-shot multiplex interaction prediction on ChEMBL.

| Setting | Model | AUROC | AUPRC | HS | SA |
|---|---|---|---|---|---|
| T | XGB | 0.515±0.013 | 0.607±0.018 | 0.699±0.007 | 0.587±0.012 |
| | MLP | 0.531±0.005 | 0.629±0.012 | 0.657±0.009 | 0.613±0.010 |
| | XGB (LLM) | 0.648±0.014 | 0.694±0.016 | 0.735±0.010 | 0.721±0.011 |
| | MLP (LLM) | 0.624±0.008 | 0.678±0.013 | 0.718±0.011 | 0.697±0.009 |
| | GCN | 0.632±0.014 | 0.700±0.008 | 0.709±0.004 | 0.641±0.012 |
| | GraphSAGE | 0.631±0.005 | 0.698±0.012 | 0.734±0.006 | 0.663±0.005 |
| | Graph Transformer | 0.644±0.015 | 0.715±0.010 | 0.741±0.005 | 0.657±0.012 |
| | DGI | 0.651±0.019 | 0.719±0.005 | 0.738±0.012 | 0.673±0.011 |
| | MultiplexSAGE | 0.653±0.007 | 0.709±0.011 | 0.712±0.005 | 0.705±0.019 |
| | DMGI | 0.661±0.017 | 0.749±0.008 | 0.773±0.021 | 0.712±0.014 |
| | HDMI | 0.663±0.012 | 0.762±0.014 | 0.789±0.012 | 0.730±0.006 |
| | R-GCN | 0.647±0.016 | 0.725±0.011 | 0.760±0.031 | 0.719±0.023 |
| | R-GAT | 0.655±0.013 | 0.742±0.009 | 0.771±0.025 | 0.722±0.010 |
| | CoSMIG | 0.661±0.021 | 0.732±0.011 | 0.755±0.009 | 0.702±0.025 |
| | DGCL | 0.659±0.014 | 0.712±0.005 | 0.739±0.007 | 0.698±0.009 |
| | **CAZI-MBN** | **0.812±0.008** | **0.863±0.006** | **0.889±0.014** | **0.757±0.011** |
| ZS | XGB | 0.481±0.023 | 0.542±0.012 | 0.608±0.014 | 0.574±0.009 |
| | MLP | 0.487±0.012 | 0.539±0.006 | 0.621±0.012 | 0.581±0.018 |
| | XGB (LLM) | 0.622±0.019 | 0.712±0.011 | 0.637±0.013 | 0.616±0.012 |
| | MLP (LLM) | 0.634±0.014 | 0.728±0.009 | 0.691±0.015 | 0.639±0.014 |
| | GCN | 0.612±0.013 | 0.702±0.007 | 0.633±0.006 | 0.607±0.011 |
| | GraphSAGE | 0.624±0.009 | 0.713±0.018 | 0.640±0.006 | 0.611±0.008 |
| | Graph Transformer | 0.628±0.011 | 0.720±0.008 | 0.725±0.013 | 0.628±0.014 |
| | DGI | 0.619±0.010 | 0.709±0.022 | 0.708±0.007 | 0.634±0.019 |
| | MultiplexSAGE | 0.638±0.007 | 0.718±0.012 | 0.711±0.008 | 0.699±0.009 |
| | DMGI | 0.652±0.021 | 0.745±0.015 | 0.756±0.007 | 0.711±0.009 |
| | HDMI | 0.648±0.015 | 0.735±0.011 | 0.726±0.004 | 0.716±0.011 |
| | R-GCN | 0.630±0.013 | 0.719±0.009 | 0.714±0.018 | 0.687±0.012 |
| | R-GAT | 0.643±0.026 | 0.731±0.012 | 0.720±0.007 | 0.708±0.017 |
| | CoSMIG | 0.641±0.011 | 0.733±0.007 | 0.729±0.010 | 0.683±0.005 |
| | DGCL | 0.631±0.019 | 0.702±0.018 | 0.693±0.007 | 0.675±0.012 |
| | **CAZI-MBN** | **0.791±0.018** | **0.839±0.015** | **0.857±0.011** | **0.723±0.009** |

Table 12: Evaluation of transductive and zero-shot multiplex interaction prediction on PINNACLE.

| Setting | Model | AUROC | AUPRC | HS | SA |
|---|---|---|---|---|---|
| T | XGB | 0.650±0.007 | 0.644±0.007 | 0.651±0.005 | 0.660±0.023 |
| | MLP | 0.676±0.008 | 0.668±0.008 | 0.656±0.011 | 0.666±0.016 |
| | XGB (LLM) | 0.742±0.010 | 0.767±0.009 | 0.744±0.007 | 0.734±0.018 |
| | MLP (LLM) | 0.733±0.011 | 0.751±0.010 | 0.699±0.010 | 0.708±0.014 |
| | GCN | 0.659±0.010 | 0.670±0.012 | 0.666±0.014 | 0.679±0.021 |
| | GraphSAGE | 0.692±0.017 | 0.677±0.008 | 0.686±0.005 | 0.693±0.025 |
| | Graph Transformer | 0.701±0.007 | 0.704±0.015 | 0.700±0.005 | 0.726±0.024 |
| | DGI | 0.688±0.009 | 0.695±0.010 | 0.698±0.006 | 0.684±0.020 |
| | MultiplexSAGE | 0.752±0.020 | 0.775±0.022 | 0.763±0.009 | 0.749±0.018 |
| | DMGI | 0.768±0.021 | 0.784±0.025 | 0.759±0.014 | 0.762±0.021 |
| | HDMI | 0.773±0.022 | 0.798±0.023 | 0.776±0.017 | 0.766±0.016 |
| | R-GCN | 0.761±0.017 | 0.786±0.012 | 0.754±0.024 | 0.753±0.019 |
| | R-GAT | 0.759±0.009 | 0.783±0.013 | 0.760±0.011 | 0.743±0.015 |
| | PIPR | 0.747±0.025 | 0.759±0.018 | 0.741±0.026 | 0.732±0.020 |
| | xCAPT5 | 0.781±0.011 | 0.804±0.014 | 0.726±0.012 | 0.752±0.015 |
| | **CAZI-MBN** | **0.831±0.018** | **0.845±0.011** | **0.751±0.009** | **0.772±0.013** |
| ZS | XGB | 0.626±0.023 | 0.628±0.010 | 0.636±0.015 | 0.639±0.026 |
| | MLP | 0.652±0.015 | 0.650±0.017 | 0.626±0.007 | 0.642±0.022 |
| | XGB (LLM) | 0.719±0.020 | 0.731±0.012 | 0.688±0.014 | 0.671±0.021 |
| | MLP (LLM) | 0.714±0.017 | 0.729±0.014 | 0.683±0.009 | 0.685±0.019 |
| | GCN | 0.639±0.017 | 0.652±0.016 | 0.649±0.012 | 0.654±0.011 |
| | GraphSAGE | 0.673±0.013 | 0.666±0.019 | 0.668±0.023 | 0.668±0.015 |
| | Graph Transformer | 0.687±0.019 | 0.690±0.017 | 0.680±0.018 | 0.705±0.019 |
| | DGI | 0.674±0.011 | 0.678±0.022 | 0.679±0.015 | 0.663±0.020 |
| | MultiplexSAGE | 0.728±0.020 | 0.753±0.019 | 0.747±0.021 | 0.725±0.020 |
| | DMGI | 0.744±0.020 | 0.762±0.024 | 0.738±0.025 | 0.738±0.022 |
| | HDMI | 0.748±0.025 | 0.776±0.021 | 0.757±0.024 | 0.742±0.014 |
| | R-GCN | 0.733±0.032 | 0.765±0.019 | 0.742±0.012 | 0.733±0.017 |
| | R-GAT | 0.726±0.023 | 0.758±0.020 | 0.734±0.025 | 0.735±0.016 |
| | PIPR | 0.721±0.011 | 0.740±0.018 | 0.716±0.019 | 0.711±0.020 |
| | xCAPT5 | 0.785±0.012 | 0.791±0.015 | 0.733±0.013 | 0.739±0.016 |
| | **CAZI-MBN** | **0.812±0.013** | **0.820±0.008** | **0.748±0.010** | **0.763±0.011** |

Table 13: Evaluation of transductive and zero-shot multiplex interaction prediction on MetaConserve.

| Setting | Model | AUROC | AUPRC | HS | SA |
|---|---|---|---|---|---|
| T | XGB | 0.611±0.006 | 0.612±0.023 | 0.617±0.021 | 0.613±0.010 |
| | MLP | 0.617±0.010 | 0.614±0.015 | 0.620±0.017 | 0.628±0.007 |
| | XGB (LLM) | 0.662±0.009 | 0.671±0.018 | 0.658±0.020 | 0.669±0.013 |
| | MLP (LLM) | 0.694±0.012 | 0.702±0.015 | 0.707±0.016 | 0.703±0.010 |
| | GCN | 0.630±0.007 | 0.642±0.022 | 0.628±0.011 | 0.633±0.018 |
| | GraphSAGE | 0.649±0.013 | 0.646±0.011 | 0.644±0.021 | 0.645±0.012 |
| | Graph Transformer | 0.663±0.022 | 0.662±0.017 | 0.653±0.012 | 0.656±0.007 |
| | DGI | 0.668±0.020 | 0.673±0.019 | 0.665±0.010 | 0.669±0.010 |
| | MultiplexSAGE | 0.693±0.012 | 0.705±0.021 | 0.678±0.018 | 0.672±0.014 |
| | DMGI | 0.717±0.008 | 0.716±0.017 | 0.719±0.015 | 0.720±0.013 |
| | HDMI | 0.726±0.015 | 0.729±0.021 | 0.715±0.006 | 0.718±0.015 |
| | R-GCN | 0.712±0.017 | 0.718±0.010 | 0.698±0.013 | 0.703±0.009 |
| | R-GAT | 0.720±0.028 | 0.723±0.017 | 0.714±0.013 | 0.710±0.014 |
| | MolTrans | 0.710±0.014 | 0.720±0.008 | 0.719±0.011 | 0.716±0.020 |
| | DrugBAN | 0.737±0.010 | 0.714±0.012 | 0.710±0.019 | 0.711±0.008 |
| | **CAZI-MBN** | **0.752±0.020** | **0.744±0.012** | **0.779±0.008** | **0.656±0.014** |
| ZS | XGB | 0.598±0.013 | 0.591±0.015 | 0.600±0.007 | 0.598±0.010 |
| | MLP | 0.607±0.016 | 0.600±0.019 | 0.612±0.010 | 0.621±0.015 |
| | XGB (LLM) | 0.631±0.014 | 0.637±0.017 | 0.635±0.009 | 0.633±0.012 |
| | MLP (LLM) | 0.659±0.018 | 0.662±0.016 | 0.664±0.012 | 0.672±0.014 |
| | GCN | 0.617±0.015 | 0.621±0.019 | 0.610±0.012 | 0.622±0.013 |
| | GraphSAGE | 0.630±0.010 | 0.633±0.016 | 0.627±0.010 | 0.634±0.006 |
| | Graph Transformer | 0.651±0.007 | 0.660±0.020 | 0.652±0.015 | 0.648±0.015 |
| | DGI | 0.646±0.018 | 0.647±0.018 | 0.638±0.014 | 0.640±0.008 |
| | MultiplexSAGE | 0.697±0.015 | 0.692±0.023 | 0.685±0.011 | 0.688±0.012 |
| | DMGI | 0.702±0.012 | 0.701±0.015 | 0.698±0.017 | 0.695±0.009 |
| | HDMI | 0.709±0.017 | 0.710±0.014 | 0.704±0.014 | 0.707±0.009 |
| | R-GCN | 0.699±0.021 | 0.702±0.012 | 0.683±0.016 | 0.685±0.010 |
| | R-GAT | 0.703±0.019 | 0.705±0.022 | 0.691±0.014 | 0.699±0.018 |
| | MolTrans | 0.688±0.007 | 0.700±0.017 | 0.690±0.007 | 0.693±0.012 |
| | DrugBAN | 0.692±0.017 | 0.705±0.020 | 0.707±0.015 | 0.708±0.011 |
| | **CAZI-MBN** | **0.738±0.015** | **0.722±0.010** | **0.749±0.020** | **0.653±0.009** |

Table 14: Evaluation of transductive and zero-shot multiplex interaction prediction on TRRUST.

| Setting | Model | AUROC | AUPRC | HS | SA |
|---------|-------|-------|-------|-----|-----|
| T | XGB | 0.786±0.006 | 0.791±0.016 | 0.794±0.005 | 0.788±0.007 |
| | MLP | 0.796±0.008 | 0.799±0.019 | 0.801±0.011 | 0.803±0.012 |
| | XGB (LLM) | 0.854±0.019 | 0.863±0.027 | 0.852±0.014 | 0.867±0.018 |
| | MLP (LLM) | 0.861±0.022 | 0.868±0.031 | 0.856±0.017 | 0.859±0.021 |
| | GCN | 0.810±0.008 | 0.815±0.022 | 0.812±0.008 | 0.817±0.005 |
| | GraphSAGE | 0.822±0.010 | 0.826±0.007 | 0.819±0.008 | 0.824±0.010 |
| | Graph Transformer | 0.832±0.005 | 0.835±0.015 | 0.837±0.004 | 0.830±0.006 |
| | DGI | 0.841±0.014 | 0.844±0.011 | 0.845±0.009 | 0.847±0.006 |
| | MultiplexSAGE | 0.860±0.018 | 0.864±0.010 | 0.858±0.009 | 0.866±0.005 |
| | DMGI | 0.871±0.013 | 0.874±0.007 | 0.869±0.008 | 0.872±0.008 |
| | HDMI | 0.878±0.008 | 0.879±0.014 | 0.883±0.005 | 0.881±0.004 |
| | R-GCN | 0.865±0.011 | 0.867±0.013 | 0.861±0.020 | 0.870±0.015 |
| | R-GAT | 0.868±0.019 | 0.871±0.014 | 0.866±0.018 | 0.875±0.023 |
| | DL-GGI | 0.857±0.006 | 0.862±0.012 | 0.854±0.007 | 0.860±0.005 |
| | GENER | 0.867±0.012 | 0.868±0.006 | 0.865±0.010 | 0.869±0.006 |
| | **CAZI-MBN** | **0.905±0.013** | **0.872±0.008** | **0.791±0.023** | **0.784±0.015** |
| ZS | XGB | 0.779±0.012 | 0.781±0.006 | 0.783±0.015 | 0.777±0.014 |
| | MLP | 0.789±0.010 | 0.790±0.014 | 0.793±0.010 | 0.794±0.006 |
| | XGB (LLM) | 0.828±0.143 | 0.836±0.198 | 0.842±0.117 | 0.831±0.022 |
| | MLP (LLM) | 0.845±0.009 | 0.852±0.017 | 0.837±0.021 | 0.851±0.015 |
| | GCN | 0.801±0.011 | 0.804±0.010 | 0.800±0.006 | 0.805±0.004 |
| | GraphSAGE | 0.812±0.006 | 0.817±0.012 | 0.809±0.009 | 0.815±0.007 |
| | Graph Transformer | 0.823±0.008 | 0.826±0.016 | 0.828±0.006 | 0.820±0.005 |
| | DGI | 0.832±0.011 | 0.835±0.014 | 0.836±0.007 | 0.838±0.006 |
| | MultiplexSAGE | 0.850±0.007 | 0.854±0.010 | 0.848±0.004 | 0.855±0.005 |
| | DMGI | 0.861±0.008 | 0.864±0.012 | 0.858±0.005 | 0.860±0.008 |
| | HDMI | 0.868±0.006 | 0.869±0.010 | 0.872±0.006 | 0.870±0.004 |
| | R-GCN | 0.854±0.013 | 0.856±0.009 | 0.850±0.017 | 0.861±0.012 |
| | R-GAT | 0.859±0.015 | 0.863±0.012 | 0.859±0.019 | 0.863±0.017 |
| | DL-GGI | 0.848±0.006 | 0.852±0.006 | 0.844±0.008 | 0.851±0.006 |
| | GENER | 0.857±0.007 | 0.858±0.008 | 0.855±0.006 | 0.859±0.007 |
| | **CAZI-MBN** | **0.899±0.017** | **0.869±0.013** | **0.775±0.019** | **0.764±0.007** |

not the main bottleneck in our design. We quantify sparsity using the multiplex rate (see Table 8), which ranges from 1.34 percent in DGIdb to 27.10 percent in MetaConserve, representing nearly a twenty-fold difference in cross-layer connectivity. Despite this wide variation, CAZI-MBN achieves stable zero-shot AUROC and AUPRC across all five benchmarks. Notably, DGIdb, the dataset with the lowest multiplex rate, still achieves strong zero-shot performance and exceeds all baseline models, indicating that the method does not rely on dense cross-layer structure. MetaConserve, which also has both a low multiplex rate and a small node–edge ratio (329 nodes and 377 edges), likewise shows robust performance. These results suggest that CAZI-MBN requires only a reasonable amount of relational signal to benefit from multiplex structure, and that the model continues to perform reliably even when the multiplex graph is relatively sparse. In practice, this implies that CAZI-MBN can operate effectively on real-world biological networks, which are commonly sparse, while still leveraging the available multiplex information to provide more informed predictions.

### E.3.2 INTERACTION TYPE-SPECIFIC ANALYSIS.

To assess performance at a more detailed level, we focus on minority interaction types across the five MBNs. Accuracy is used as the evaluation metric, and CAZI-MBN is compared against the top-performing benchmark in each category (sequence-based, single-graph-based, multiplex-graph-based, and domain-specific), based on their accuracy for each minority interaction type.

Figure 5 presents the interaction type-specific prediction accuracy across five datasets in the zero-shot setting. CAZI-MBN consistently outperforms the best benchmark models, demonstrating strong capability in capturing subtle class distinctions and reinforcing its suitability for zero-shot prediction in complex biological networks. This is particularly important in practical applications where rare or underrepresented interaction types may be biologically significant.

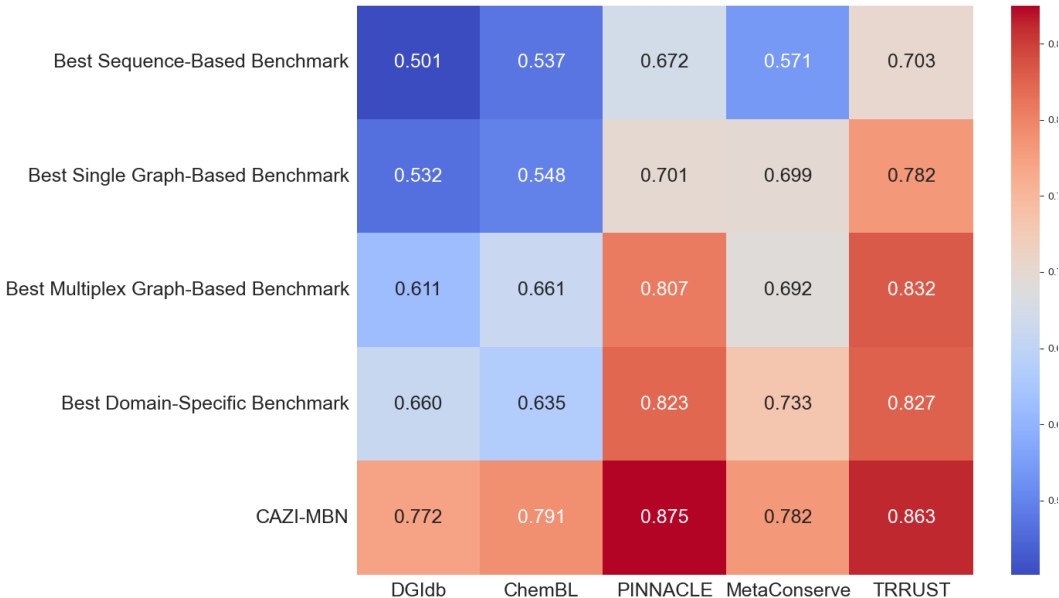

Figure 5: Interaction type-specific performance analysis across five datasets, evaluated by prediction accuracy.

### E.3.3 ABLATION STUDIES

To rigorously evaluate the individual contributions of each core component in the CAZI-MBN framework, we conducted systematic ablation studies by selectively removing one module at a time while keeping the rest of the architecture intact. This approach allows us to isolate the functional impact of

each component on overall model performance and to better understand how each contributes to the model's capacity to learn from multiplex biological interaction data.

Figure 6 shows the module-wise performance drops across all datasets under both transductive and zero-shot settings, with error bars illustrating the variance across interaction types. A summary of the ablation configurations is provided in Table 15. The detailed results are reported in Tables 16 and 17.

Ablation studies further validate the necessity of CAZI-MBN's modular architecture. Across all datasets, CAZI-MBN consistently outperforms every ablated variant, and the most significant performance drops occur when the pre-trained LLMs or the CAE module are removed. The LLMs capture fine-grained biochemical semantics across drugs, genes, and proteins, while the CAE module introduces multiplex-aware, context-sensitive attention and contrastive alignment that enhance cross-layer consistency. Together, these modules form the backbone of CAZI-MBN's representational power, enabling it to jointly model both sequence information and heterogeneous interaction patterns.

CAZI-MBN was specifically designed to integrate high-quality biochemical sequence embeddings with multiplex graph structure and to distill this joint information for generalization. This design choice is particularly critical in the zero-shot setting, where models must predict interactions for entities with no prior neighborhood context. The strong contribution of LLM-based embeddings to overall performance is therefore not surprising, as they provide rich sequence-level semantics that directly support generalization. Indeed, when compared against baselines that lack domain-specific LLMs, CAZI-MBN shows substantial improvements in zero-shot prediction, confirming the importance of leveraging modality-specific pre-trained models.

Taken together, these findings demonstrate that CAZI-MBN's strength lies in its capacity to integrate complementary modalities, sequence-level semantics and relational structure, into coherent representations. This synergy allows the framework to capture both fine-grained molecular details and higher-order dependencies across interaction layers, yielding robust performance in transductive tasks and strong generalization in zero-shot scenarios.

Table 15: Summary of ablation variants.

| Variant | Description |
|---|---|
| w/o UGT | Remove UGT; use only LLM-derived features as input |
| w/o LLMs | Replace LLM features with 6-mer and Morgan fingerprints |
| w/o CAE | Replace CAE with a single GCN per interaction type |
| w/o MoE | Replace MoE with a plain MLP per interaction type |
| w/o $L_{dis}$ | Discard discriminator |
| w/o $L_{reg}$ | Discard consensus regularizer and replace with attention-only fusion for Z |

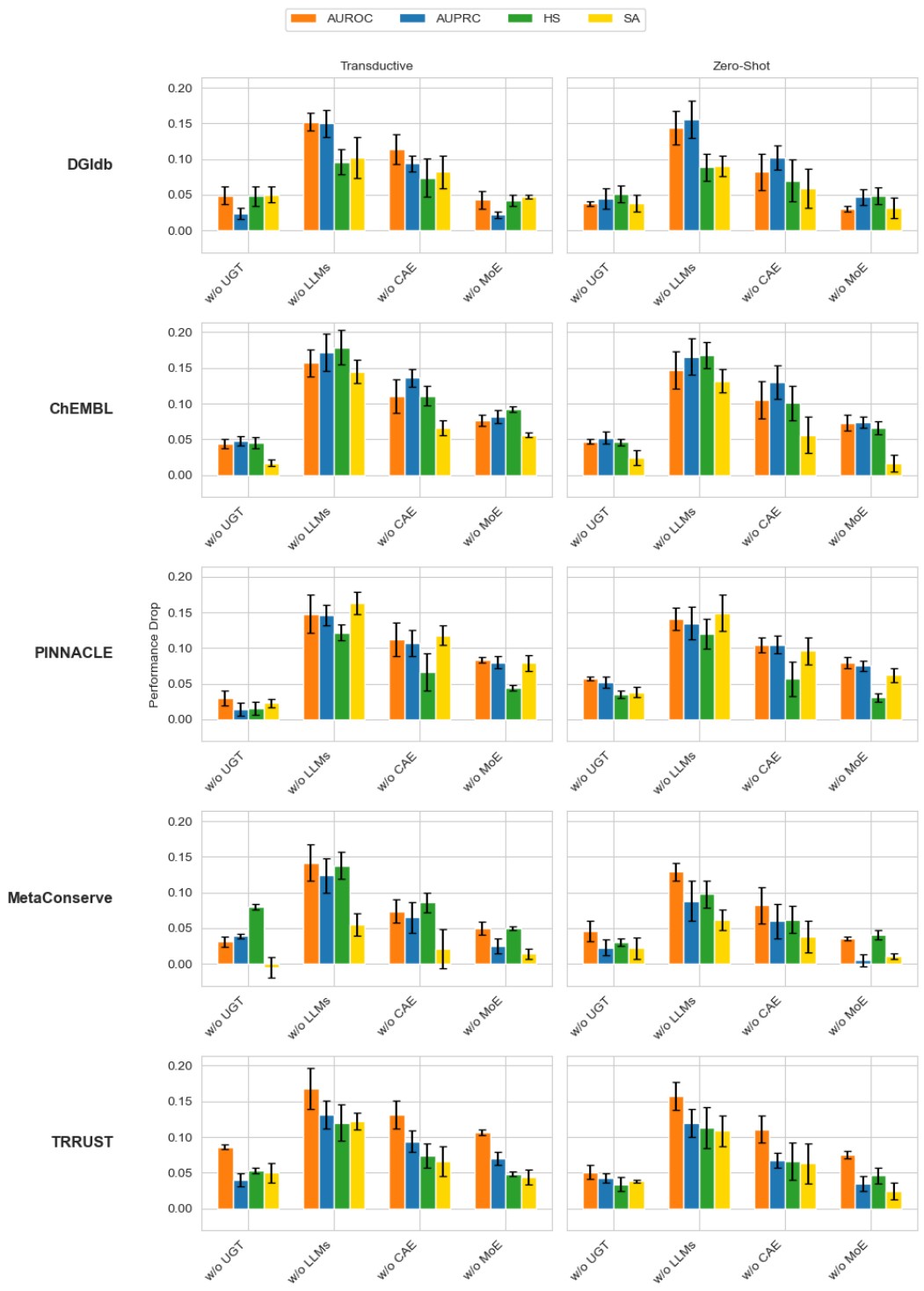

Figure 6: Module-wise ablation analysis across five MBNs under transductive and zero-shot settings. Bars show the performance drop for four evaluation metrics when removing each module (UGT, LLMs, CAE, MoE). Error bars reflect the variance across interaction types.

Table 16: Ablation results in transductive setting across five MBNs.

| Dataset | Model | AUROC | AUPRC | HS | SA |
|---|---|---|---|---|---|
| DGIdb | w/o UGT | 0.666±0.015 | 0.705±0.008 | 0.639±0.012 | 0.634±0.017 |
| | w/o LLMs | 0.563±0.021 | 0.579±0.023 | 0.591±0.014 | 0.582±0.016 |
| | w/o CAE | 0.601±0.014 | 0.635±0.018 | 0.613±0.016 | 0.602±0.012 |
| | w/o MoE | 0.672±0.019 | 0.707±0.011 | 0.645±0.020 | 0.637±0.018 |
| | w/o $L_{dis}$ | 0.692±0.024 | 0.701±0.017 | 0.673±0.019 | 0.655±0.022 |
| | w/o $L_{reg}$ | 0.679±0.012 | 0.696±0.013 | 0.660±0.010 | 0.644±0.014 |
| | **CAZI-MBN** | **0.715±0.007** | **0.729±0.009** | **0.687±0.011** | **0.684±0.015** |
| ChEMBL | w/o UGT | 0.768±0.010 | 0.815±0.006 | 0.844±0.010 | 0.740±0.013 |
| | w/o LLMs | 0.655±0.016 | 0.691±0.020 | 0.710±0.013 | 0.612±0.019 |
| | w/o CAE | 0.702±0.008 | 0.727±0.012 | 0.778±0.011 | 0.691±0.016 |
| | w/o MoE | 0.735±0.014 | 0.781±0.017 | 0.797±0.009 | 0.701±0.009 |
| | w/o $L_{dis}$ | 0.790±0.023 | 0.824±0.017 | 0.859±0.011 | 0.728±0.012 |
| | w/o $L_{reg}$ | 0.731±0.014 | 0.811±0.015 | 0.826±0.012 | 0.715±0.016 |
| | **CAZI-MBN** | **0.812±0.008** | **0.863±0.006** | **0.889±0.014** | **0.757±0.011** |
| PINNACLE | w/o UGT | 0.801±0.009 | 0.831±0.013 | 0.735±0.011 | 0.749±0.012 |
| | w/o LLMs | 0.683±0.019 | 0.699±0.017 | 0.629±0.019 | 0.609±0.018 |
| | w/o CAE | 0.719±0.014 | 0.738±0.015 | 0.685±0.008 | 0.654±0.014 |
| | w/o MoE | 0.748±0.015 | 0.765±0.016 | 0.707±0.010 | 0.693±0.013 |
| | w/o $L_{dis}$ | 0.811±0.016 | 0.832±0.25 | 0.740±0.013 | 0.742±0.018 |
| | w/o $L_{reg}$ | 0.795±0.011 | 0.818±0.013 | 0.737±0.013 | 0.722±0.013 |
| | **CAZI-MBN** | **0.831±0.018** | **0.845±0.011** | **0.751±0.009** | **0.772±0.013** |
| MetaConserve | w/o UGT | 0.721±0.015 | 0.705±0.010 | 0.699±0.012 | 0.661±0.011 |
| | w/o LLMs | 0.610±0.018 | 0.620±0.022 | 0.641±0.020 | 0.601±0.021 |
| | w/o CAE | 0.678±0.019 | 0.679±0.018 | 0.693±0.010 | 0.635±0.014 |
| | w/o MoE | 0.702±0.016 | 0.719±0.020 | 0.729±0.008 | 0.642±0.010 |
| | w/o $L_{dis}$ | 0.731±0.011 | 0.720±0.009 | 0.737±0.014 | 0.625±0.018 |
| | w/o $L_{reg}$ | 0.727±0.015 | 0.719±0.014 | 0.743±0.011 | 0.637±0.014 |
| | **CAZI-MBN** | **0.752±0.020** | **0.744±0.012** | **0.779±0.008** | **0.656±0.014** |
| TRRUST | w/o UGT | 0.813±0.013 | 0.829±0.011 | 0.722±0.010 | 0.714±0.015 |
| | w/o LLMs | 0.731±0.024 | 0.738±0.015 | 0.655±0.012 | 0.642±0.018 |
| | w/o CAE | 0.768±0.017 | 0.775±0.020 | 0.701±0.013 | 0.698±0.011 |
| | w/o MoE | 0.792±0.014 | 0.799±0.016 | 0.727±0.019 | 0.720±0.012 |
| | w/o $L_{dis}$ | 0.857±0.009 | 0.812±0.014 | 0.733±0.013 | 0.730±0.020 |
| | w/o $L_{reg}$ | 0.831±0.012 | 0.840±0.012 | 0.728±0.011 | 0.722±0.013 |
| | **CAZI-MBN** | **0.899±0.017** | **0.869±0.013** | **0.775±0.019** | **0.764±0.007** |

Table 17: Ablation results in zero-shot setting across five MBNs for CAZI-MBN(KD).

| Dataset | Model | AUROC | AUPRC | HS | SA |
|---|---|---|---|---|---|
| DGIdb | w/o UGT | 0.633±0.015 | 0.664±0.010 | 0.637±0.012 | 0.625±0.014 |
| | w/o LLMs | 0.527±0.017 | 0.553±0.025 | 0.599±0.018 | 0.573±0.015 |
| | w/o CAE | 0.589±0.019 | 0.607±0.020 | 0.618±0.009 | 0.604±0.012 |
| | w/o MoE | 0.641±0.012 | 0.662±0.015 | 0.639±0.011 | 0.631±0.011 |
| | w/o $L_{dis}$ | 0.643±0.017 | 0.677±0.009 | 0.819±0.022 | 0.690±0.015 |
| | w/o $L_{reg}$ | 0.593±0.012 | 0.616±0.013 | 0.621±0.011 | 0.601±0.012 |
| | **CAZI-MBN** | **0.671±0.008** | **0.709±0.011** | **0.688±0.009** | **0.663±0.014** |
| ChEMBL | w/o UGT | 0.744±0.016 | 0.787±0.014 | 0.811±0.010 | 0.699±0.017 |
| | w/o LLMs | 0.644±0.021 | 0.673±0.027 | 0.689±0.014 | 0.591±0.015 |
| | w/o CAE | 0.686±0.017 | 0.709±0.016 | 0.756±0.008 | 0.667±0.011 |
| | w/o MoE | 0.718±0.010 | 0.765±0.012 | 0.791±0.011 | 0.707±0.013 |
| | w/o $L_{dis}$ | 0.772±0.023 | 0.810±0.019 | 0.815±0.017 | 0.693±0.025 |
| | w/o $L_{reg}$ | 0.718±0.013 | 0.746±0.014 | 0.775±0.010 | 0.686±0.014 |
| | **CAZI-MBN** | **0.791±0.018** | **0.839±0.015** | **0.857±0.011** | **0.723±0.009** |
| PINNACLE | w/o UGT | 0.755±0.012 | 0.768±0.018 | 0.713±0.014 | 0.725±0.012 |
| | w/o LLMs | 0.671±0.019 | 0.685±0.021 | 0.628±0.012 | 0.614±0.014 |
| | w/o CAE | 0.708±0.016 | 0.715±0.014 | 0.691±0.010 | 0.667±0.016 |
| | w/o MoE | 0.733±0.009 | 0.745±0.011 | 0.717±0.007 | 0.701±0.010 |
| | w/o $L_{dis}$ | 0.775±0.015 | 0.801±0.008 | 0.715±0.021 | 0.730±0.019 |
| | w/o $L_{reg}$ | 0.758±0.011 | 0.794±0.014 | 0.732±0.012 | 0.712±0.013 |
| | **CAZI-MBN** | **0.812±0.013** | **0.820±0.008** | **0.748±0.010** | **0.763±0.011** |
| MetaConserve | w/o UGT | 0.692±0.011 | 0.699±0.013 | 0.719±0.012 | 0.631±0.014 |
| | w/o LLMs | 0.609±0.017 | 0.634±0.015 | 0.651±0.016 | 0.591±0.012 |
| | w/o CAE | 0.656±0.014 | 0.662±0.016 | 0.687±0.009 | 0.615±0.011 |
| | w/o MoE | 0.703±0.015 | 0.717±0.018 | 0.708±0.011 | 0.642±0.009 |
| | w/o $L_{dis}$ | 0.719±0.022 | 0.693±0.025 | 0.716±0.014 | 0.633±0.016 |
| | w/o $L_{reg}$ | 0.713±0.014 | 0.700±0.015 | 0.735±0.011 | 0.629±0.013 |
| | **CAZI-MBN** | **0.738±0.015** | **0.722±0.010** | **0.749±0.020** | **0.653±0.009** |
| TRRUST | w/o UGT | 0.848±0.013 | 0.826±0.016 | 0.741±0.015 | 0.726±0.012 |
| | w/o LLMs | 0.742±0.018 | 0.749±0.021 | 0.662±0.010 | 0.655±0.014 |
| | w/o CAE | 0.788±0.019 | 0.802±0.013 | 0.709±0.014 | 0.701±0.013 |
| | w/o MoE | 0.824±0.014 | 0.834±0.015 | 0.729±0.013 | 0.740±0.015 |
| | w/o $L_{dis}$ | 0.867±0.019 | 0.831±0.029 | 0.732±0.020 | 0.735±0.011 |
| | w/o $L_{reg}$ | 0.851±0.012 | 0.818±0.011 | 0.729±0.013 | 0.716±0.014 |
| | **CAZI-MBN** | **0.899±0.017** | **0.869±0.013** | **0.775±0.019** | **0.764±0.007** |

### E.3.4 CASE STUDY: INTERACTIONS INVOLVING IBD-RELATED GENES AND PROTEINS

To evaluate the effectiveness of our approach in real-world biological scenarios, we conducted a case study focused on zero-shot multiplex interaction prediction involving IBD-related genes and their protein products. The selected genes, *GAPDH*, *PC*, *RPS14*, *OAT*, and *NDUFAB1*, were chosen based on their established functional relevance to IBD and their evolutionary conservation from *E. coli* orthologs, including *gapA*, *accC*, *rpsK*, *orn*, and *fabD*. These genes represent a diverse set of biological functions, such as energy metabolism, redox balance, immune regulation, and mitochondrial activity, all of which are closely associated with IBD pathogenesis. This focused selection enables a robust evaluation of the model's ability to predict novel, biologically meaningful interactions involving entirely unseen entities.

The genes and their IBD-relevant roles are summarized as follows:

- *GAPDH* (ortholog: *gapA*): Known for its central role in glycolysis, GAPDH also regulates T-cell function and cytokine production, highlighting its dual role in metabolism and immune modulation (Sheng & Wang, 2009; Bas et al., 2004; Barber et al., 2005).

- *PC* (ortholog: *accC*): Pyruvate carboxylase plays a key role in gluconeogenesis and epithelial energy metabolism, both of which are affected in IBD (Algieri et al., 2016; Ferrer-Picón et al., 2020; Tang et al., 2015; Bos & Laukens, 2020).

- *RPS14* (ortholog: *rpsK*): A ribosomal protein with emerging evidence linking it to immune and stress response regulation in intestinal inflammation (Sallman & List, 2019; Zou & Zhang, 2021; Lin et al., 2020; Das et al., 2024).

- *OAT* (ortholog: *orn*): Ornithine aminotransferase regulates amino acid metabolism and redox balance, with links to epithelial injury in IBD (Ji et al., 2023; Smith et al., 2021; Lan et al., 2023; Gobert et al., 2004).

- *NDUFAB1* (ortholog: *fabD*): This mitochondrial protein supports fatty acid synthesis and electron transport and is associated with epithelial barrier integrity (Guerbette et al., 2022; Rath et al., 2018; Kim et al., 2021).

For each dataset-specific case study, a separate model was trained using a standard zero-shot setting. All biological entities and interactions involving the five specified genes and their protein products were held out as the test set. The remaining interactions and associated entities were randomly split into training and validation sets using an 5:1 ratio. This setup ensures that the model does not encounter the held-out genes or their associated links during training, enabling a clear evaluation of its zero-shot generalization ability.

Table 18: Per-Gene/Protein Product Statistics Across Datasets

| | DGIdb | TRRUST | PINNACLE | MetaConserve |
|---|---|---|---|---|
| *GAPDH*(*gapA*) | 34 / **41** (82.9%) | 2 / **3** (66.7%) | 103 / **140** (73.6%) | – |
| *PC*(*accC*) | 2 / **2** (100%) | 2 / **2** (100%) | – | – |
| *RPS14*(*rpsK*) | 3 / **5** (60%) | – | 340 / **454** (74.9%) | – |
| *OAT*(*orn*) | 1 / **1** (100%) | 2 / **2** (100%) | 11 / **16** (68.7%) | – |
| *NDUFAB1*(*fabD*) | 3 / **3** (100%) | – | 46 / **57** (80.7%) | 5 / **8** (62.5%) |
| **Total** | 43 / **52** (82.7%) | 6 / **7** (85.7%) | 500 / **666** (75.1%) | 5 / **8** (62.5%) |

[a] Each cell shows: *Predicted* / **Actual** (Accuracy), where *Predicted* indicates the number of interactions correctly predicted by the model, and **Actual** indicates the total number of known interactions involving the given gene or its product. "–" indicates no active interaction with the gene or its product was found in that dataset.

[b] Total interaction counts may not equal the sum of individual gene or gene product counts, as some interactions involve multiple listed genes or their products.

Table 18 presents the zero-shot prediction performance of CAZI-MBN across four benchmark datasets, reporting an overall, as well as per-gene and gene product-level interaction statistics. These results evaluate the model's ability to generalize to previously unseen entities, specifically those involving five held-out human genes and their bacterial orthologs. None of these genes or their

products were included in the training set, allowing for a rigorous assessment of the model's zero-shot capabilities in the context of sparse and novel biological interactions.

CAZI-MBN recovered a substantial proportion of known interactions across all evaluated datasets, achieving 82.7% accuracy in DGIdb, 85.7% in TRRUST, 75.1% in PINNACLE, and 62.5% in Meta-Conserve. These datasets collectively span a range of biological interaction types, including drug-gene interactions (DGIdb), transcriptional regulatory links (TRRUST), cell-type-specific protein-protein interactions (PINNACLE), and evolutionarily conserved co-occurrence patterns (MetaConserve). The model's consistently strong performance across these diverse settings underscores its robustness and flexibility in handling heterogeneous biological data.

Closer inspection of individual genes reveals that CAZI-MBN maintains strong predictive performance even for held-out genes with relatively few known interactions in the test sets. For example, *PC (accC)* and *OAT (orn)* each exhibited low interaction frequency within DGIdb and TRRUST, yet the model successfully recovered all known interactions for both cases in these datasets. This is noteworthy given that these genes were entirely excluded from training, highlighting the model's ability to generalize to novel genes and their interaction profiles. Although these results reflect only test-time performance, they demonstrate that CAZI-MBN can effectively capture biological relationships across different interaction types, including drug-gene interactions, transcriptional regulation, cell-type specific protein-protein interactions, without relying on prior exposure to the specific biological entities involved. Such flexibility is critical for modeling MBNs, where each biological entity may participate in diverse biological processes and be subject to distinct regulatory mechanisms. Overall, these findings highlight the strength of CAZI-MBN in generalizing across both interaction types and gene contexts. Its capacity to make accurate predictions in a zero-shot setting suggests its potential as a valuable tool for expanding known interaction networks, prioritizing novel gene candidates, and supporting downstream biological discovery.

Several high-confidence predictions made by CAZI-MBN are further supported by existing biomedical literature, reinforcing the biological plausibility of the model's outputs. For example, the model predicts an interaction between *GAPDH* and Omigapil (DGIdb, pred: 0.912), which is validated by evidence showing that Omigapil inhibits the GAPDH–Siah1 apoptotic signaling pathway in neuromuscular disorders (Foley et al., 2024; Zhou et al., 2015). Additionally, GAPDH has been shown to chaperone ribosomal protein L13a, stabilizing the GAIT complex and modulating inflammation-related translation, which aligns with predicted interactions between GAPDH and ribosomal proteins in PINNACLE (Jia et al., 2012). The model also identifies numerous interactions involving *RPS14* with other ribosomal proteins (e.g., RPL19, RPS17), which are consistent with its role in ribosome assembly and immune-related translational control (Zhou et al., 2015; Chan et al., 2018; Wang et al., 2022). These literature-backed findings confirm that CAZI-MBN is not only capable of recovering known links but also makes biologically coherent predictions in a fully zero-shot setting.

This case study highlights the effectiveness of multiplex-aware zero-shot learning frameworks in predicting biologically meaningful interactions for novel or under-characterized entities. It reinforces CAZI-MBN's potential as a tool for accelerating hypothesis generation and advancing discovery across complex, multilayered biological systems.

### E.3.5 PARAMETER COUNTS AND GPU TIME

Table 19 reports the dataset sizes, parameter counts, average GPU time, and average runtime per epoch. Because early stopping is used, the total number of training epochs varies across runs. Overall, the student model is substantially smaller and trains much faster than the teacher model. In practice, the student requires only a small share of the total GPU time because it has far fewer parameters and does not perform the heavier graph operations. A notable trend is that teacher runtime grows more with the number of interaction types than with the number of nodes, since a separate graph transformer is instantiated for each interaction type.

### E.3.6 TTRUST: POST-HOC PER-LABEL CALIBRATION

To assess whether a more flexible thresholding strategy could improve the multi-label metrics for datasets which lags on HS and SA, we conducted a post-hoc per-label calibration experiment on TRRUST. For each interaction type, we swept thresholds from 0.2 to 0.8 in increments of 0.1 and selected the value that maximized validation accuracy for each interaction type. This allows each

Table 19: Runtime and parameter statistics across datasets for teacher and student models.

|  | MetaConserve | DGIdb | TRRUST | PINNACLE | ChEMBL |
|---|---|---|---|---|---|
| # Nodes | 329 | 1,846 | 2,862 | 7,044 | 9,368 |
| # Interaction Types | 4 | 5 | 3 | 12 | 3 |
| Teacher Total GPU Hours | 0.463 h | 3.191 h | 2.137 h | 31.34 h | 8.456 h |
| Teacher Time/Epoch | 0.574 s | 4.332 s | 3.903 s | 115.037 s | 43.201 s |
| Student Total GPU Hours | 0.00109 h | 0.0567 h | 0.0098 h | 0.4275 h | 0.114 h |
| Student Time/Epoch | 0.005 s | 0.129 s | 0.078 s | 3.802 s | 1.311 s |
| Teacher Parameters | 2016.820 M | 3430.584 M | 2062.506 M | 6045.610 M | 2068.336 M |
| Student Parameters | 1.189 M | 1.734 M | 1.041 M | 3.568 M | 1.041 M |

label to adopt a threshold better aligned with its own score distribution. The same post-hoc per-label calibration strategy is also applied on the baselines to ensure fair comparison.

Table 20 shows the evaluation results of transductive and zero-shot multiplex interaction prediction on TRRUST after post-hoc per-label calibration. After applying per-label calibration, we find that most models achieve higher HS and SA, with CAZI-MBN showing a significant improvement on TRRUST as well. This indicates that more flexible thresholding can better accommodate differences in label prevalence and score distributions across interaction types. Allowing each label to use its own decision boundary provides a better match to this heterogeneity, suggesting that more adaptive thresholding strategies may further enhance multi-label evaluation in multiplex interaction prediction.

### E.3.7 LAYER-SPECIFIC ATTENTION WEIGHTS

In CAE, layer-level context-aware consensus attention aggregates the layer-specific node features into a shared representation $H$ and its negative counterpart $\tilde{H}$. A contrastive consensus regularizer then learns a multiplex-wide consensus embedding $Z$ by maximizing agreement with $H$ and minimizing disagreement with $\tilde{H}$, enabling the model to form a layer-informed representation.

Figure 7 shows the average layer-level attention weight distributions for all five datasets. TRRUST, ChEMBL, DGIdb, and MetaConserve each have one or two interaction types that receive most of the weight, which indicates that the model relies on a small set of strong signals in these datasets. PINNACLE contains more interaction types, which naturally reduces the variability of each individual weight, although some differences in emphasis are still visible. These patterns support the purpose of using attention weights: different layers contribute uneven amounts of signal due to variation in noise levels, edge density, and biological relevance. A fixed or uniform weighting scheme implicitly assumes that all layers are equally informative, which is often violated in multiplex biological networks. Learning attention weights allows the model to adaptively modulate each layer's influence by down-weighting noisy or weakly aligned layers and emphasizing those with stronger structural consistency. This produces a consensus embedding that better captures the dominant cross-layer relational patterns.

Table 20: Evaluation of transductive and zero-shot multiplex interaction prediction on TRRUST (Post-Hoc Per-Label Calibration).

| Setting | Model | AUROC | AUPRC | HS | SA |
|---|---|---|---|---|---|
| T | XGB | 0.786±0.006 | 0.791±0.016 | 0.788±0.017 | 0.792±0.014 |
| | MLP | 0.796±0.008 | 0.799±0.019 | 0.809±0.013 | 0.812±0.018 |
| | GCN | 0.810±0.008 | 0.815±0.022 | 0.821±0.011 | 0.826±0.010 |
| | GraphSAGE | 0.822±0.010 | 0.826±0.007 | 0.824±0.014 | 0.829±0.015 |
| | Graph Transformer | 0.832±0.005 | 0.835±0.015 | 0.841±0.010 | 0.834±0.012 |
| | DGI | 0.841±0.014 | 0.844±0.011 | 0.851±0.013 | 0.853±0.011 |
| | MultiplexSAGE | 0.860±0.018 | 0.864±0.010 | 0.868±0.012 | 0.872±0.011 |
| | DMGI | 0.871±0.013 | 0.874±0.007 | 0.875±0.014 | 0.878±0.016 |
| | HDMI | 0.878±0.008 | 0.879±0.014 | 0.887±0.009 | 0.884±0.008 |
| | R-GCN | 0.865±0.011 | 0.867±0.013 | 0.872±0.018 | 0.881±0.021 |
| | R-GAT | 0.868±0.019 | 0.871±0.014 | 0.878±0.020 | 0.885±0.017 |
| | DL-GGI | 0.857±0.006 | 0.862±0.012 | 0.863±0.013 | 0.868±0.014 |
| | GENER | 0.867±0.012 | 0.868±0.006 | 0.873±0.011 | 0.876±0.010 |
| | **CAZI-MBN (Per-Label Calibration)** | **0.905±0.013** | **0.872±0.008** | **0.886±0.015** | **0.891±0.014** |
| | **CAZI-MBN (Unified Threshold)** | **0.905±0.013** | **0.872±0.008** | **0.791±0.023** | **0.784±0.015** |
| ZS | XGB | 0.779±0.012 | 0.781±0.006 | 0.783±0.018 | 0.789±0.014 |
| | MLP | 0.789±0.010 | 0.790±0.014 | 0.799±0.020 | 0.801±0.017 |
| | GCN | 0.801±0.011 | 0.804±0.010 | 0.812±0.014 | 0.815±0.013 |
| | GraphSAGE | 0.812±0.006 | 0.817±0.012 | 0.818±0.017 | 0.821±0.016 |
| | Graph Transformer | 0.823±0.008 | 0.826±0.016 | 0.837±0.020 | 0.833±0.018 |
| | DGI | 0.832±0.011 | 0.835±0.014 | 0.848±0.018 | 0.849±0.015 |
| | MultiplexSAGE | 0.850±0.007 | 0.854±0.010 | 0.853±0.016 | 0.858±0.014 |
| | DMGI | 0.861±0.008 | 0.864±0.012 | 0.860±0.015 | 0.869±0.017 |
| | HDMI | 0.868±0.006 | 0.869±0.010 | 0.884±0.019 | 0.882±0.020 |
| | R-GCN | 0.854±0.013 | 0.856±0.009 | 0.862±0.022 | 0.870±0.021 |
| | R-GAT | 0.859±0.015 | 0.863±0.012 | 0.869±0.018 | 0.872±0.017 |
| | DL-GGI | 0.848±0.006 | 0.852±0.006 | 0.860±0.020 | 0.863±0.016 |
| | GENER | 0.857±0.007 | 0.858±0.008 | 0.858±0.021 | 0.862±0.018 |
| | **CAZI-MBN (Per-Label Calibration)** | **0.899±0.017** | **0.869±0.013** | **0.882±0.013** | **0.884±0.020** |
| | **CAZI-MBN (Unified Threshold)** | **0.899±0.017** | **0.869±0.013** | **0.775±0.019** | **0.764±0.007** |

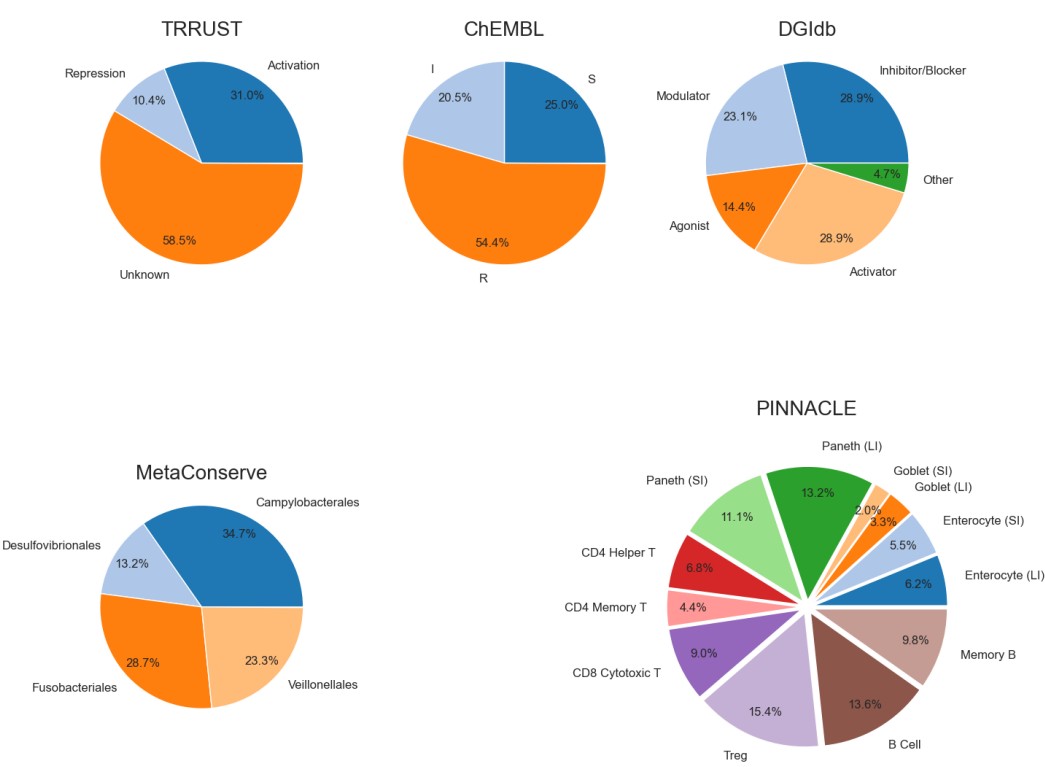

Figure 7: Attention weight distributions across all five multiplex biological datasets. Each pie chart shows the relative importance assigned to different interaction types within a dataset.

