# OpenReview forum: "Distilling and Adapting:  A Topology-Aware Framework for Zero-Shot Interaction Prediction in Multiplex Biological Networks"
_ICLR.cc/2026/Conference — ICLR 2026 Poster_

### Official Review · Reviewer_EFYv · 2025-10-29

**Soundness:** 3
**Presentation:** 3
**Contribution:** 2
**Rating:** 6
**Confidence:** 3

**Summary:**

In this study, the authors proposed a CAZI-MBN (Context-Aware and Zero-shot Interaction prediction in Multiplex Biological Networks) model framework for multiplex biological interaction prediction. The evaluation results showed improved performance.

**Strengths:**

The CAZI-MBN (Context-Aware and Zero-shot Interaction prediction in Multiplex Biological Networks) model framework for multiplex biological interaction prediction is kind of new.
The evaluation results showed improved performance.

**Weaknesses:**

For the interactome data, there are much more negative data (no interactions). It is unclear how was the negative data selected and evaluated using the proposed model.

**Questions:**

It is unclear how many negative data was selected for the model evaluation?
For the zero-shot setting, will the performance heavliy based on the density of the available positive controls?
In the datasets, it is unclear how many interactions (in addition to the no. of nodes)

---

> ### Author Response · Authors · 2025-11-25
> **Rebuttal to Reviewer EFYv - Part 1**
>
> Dear Reviewer,
>
> We thank you for acknowledging the novelty and performance of our model. CAZI-MBN integrates state-of-the-art domain-specific LLMs, multiplex modeling, and an MoE for multi-label prediction. We also provide extensive empirical evaluation, including interaction-type-specific analyses, ablations, and case studies to validate the model’s performance.
>
> Below is our response to your questions.
>
> **W1/Q1: Negative Sampling**
>
> In multiplex networks, negative sampling is defined over node pairs rather than individual layers. We treat a node pair $(u,v)$ as a positive instance if it appears in at least one layer, and as a negative instance if it does not appear in any layer, corresponding to an all-zero label vector. We use a standard random negative sampling strategy as in R-GCN [1] and SuperGAT [2]. Our ratio is 1:1 , which means that each positive node pair is matched with one randomly selected non-interacting pair drawn from the set of all unobserved pairs.
>
> We have also included this in our updated manuscript in **Supplementary E.2**
>
> ---
>
> **Q2: On Graph Density and Zero-Shot Performance**
>
> There is naturally some impact: if the multiplex graph is denser, the teacher can extract richer structural signals, which can strengthen the distilled representations. Sparsity can limit the amount of structure the teacher learns, but it is not the main bottleneck in our design.
>
> To measure the structural sparsity across layers in multiplex networks, we can use the **multiplex rate**:
>
> **Multiplex Rate = ( # node pairs that appear in ≥2 layers ) / ( # unique node pairs in the multiplex graph ) × 100%.**
>
>
>
> Below is a table that summarizes the multiplex rate of our datasets:
> | **Dataset**     | **Multiplex Rate (%)** |
> |-----------------|------------------------|
> | DGIdb           | 1.34                   |
> | TRRUST          | 8.90                   |
> | ChEMBL          | 14.45                  |
> | PINNACLE        | 23.26                  |
> | MetaConserve    | 27.10                  |
>
> Despite this ~20× variation in multiplex rate, **zero-shot AUROC/AUPRC remain stable across datasets**. Notably, DGIdb, the dataset with the lowest multiplex rate, still achieves strong zero-shot performance and outperforms all baseline models. MetaConserve, which has a very lower node–edge ratio (329 nodes and 377 edges, which suggests that this network is rather sparse), also shows robust performance in zero-shot setting. This suggests that the method requires only a reasonable amount of relational signal to effectively benefit from the available graph structure. **Even when the graph is relatively sparse, the addition of structural information is still helpful.** It's simply more or less information for the model to learn from, rather than something that actively and significantly hurts performance.
>
> This analysis has also been updated to our revised manuscript, in **Supplementary Section E.3.1**.

---

> ### Author Response · Authors · 2025-11-25
> **Rebuttal to Reviewer EFYv - Part 2**
>
> **Q3: Summary Statistics of Datasets**
>
> Below is the summary statistics of our five datasets. You can also find this in the updated **Supplementary Section D (Table 8)**.
>
> | Dataset        | Interaction Types                                                                                                      | # Nodes | # Edges   | # Interaction Types | Multiplex Rate (%) |
> |----------------|-------------------------------------------------------------------------------------------------------------------------------|---------|-----------|----------------------|---------------------|
> | **DGIdb**      | Inhibitor (7,004); Modulator (1,439); Agonist (915); Activator (104); Other (170)                                            | 1,846   | 8,443     | 5                    | 1.34                |
> | **ChEMBL**     | S (120,265); I (33,925); R (94,392)                                                                                           | 9,368   | 248,582   | 3                    | 14.45               |
> | **PINNACLE**   | Enterocyte-LI (4,089); Enterocyte-SI (7,433); Goblet-LI (3,073); Goblet-SI (6,258); Paneth-LI (8,291); Paneth-SI (15,093); CD4-Helper (6,893); CD4-Memory (7,190); CD8-Cytotoxic (8,172); Treg (15,626); B (9,261); Memory-B (10,064) | 7,044   | 101,443   | 12                   | 23.26               |
> | **MetaConserve** | Campylobacterales (109); Desulfovibrionales (98); Fusobacteriales (94); Veillonellales (76)                                | 329     | 377       | 4                    | 27.10               |
> | **TRRUST**     | Activation (3,149); Repression (1,922); Unknown (4,325)                                                                       | 2,862   | 9,396     | 3                    | 8.90                |
>
> ---
>
> Again, we really appreciate your review and questions regarding negative sampling, graph sparsity, and dataset statistics. We hope our response address your concerns.
>
> Thank again,
>
> Authors
>
> ---
>
> References:
>
> [1] Schlichtkrull, Michael, et al. "Modeling relational data with graph convolutional networks." European semantic web conference. Cham: Springer International Publishing, 2018.
>
> [2] Kim, Dongkwan, and Alice Oh. "How to find your friendly neighborhood: Graph attention design with self-supervision." arXiv preprint arXiv:2204.04879 (2022).

---

> > ### Author Response · Authors · 2025-11-28
> > **Hope You Can Take a Look at Our Responses**
> >
> > Dear Reviewer,
> > We hope this message finds you well. As the discussion period is approaching its end, we wanted to kindly draw your attention to our responses to your valuable comments. We hope they address your concerns, and we would greatly appreciate any additional thoughts or clarifications you may have.
> >
> > Your feedback is invaluable, and we are eager to refine our work based on your insights. If there are any remaining points you would like us to consider, please feel free to let us know.
> >
> > Thank you very much for your time and effort in reviewing our paper.
> >
> > The authors

---

### Official Review · Reviewer_Xr4R · 2025-10-30

**Soundness:** 2
**Presentation:** 3
**Contribution:** 2
**Rating:** 4
**Confidence:** 4

**Summary:**

This paper tackles zero-shot interaction prediction in multiplex biological networks by combining domain-specific sequence embeddings with a topology-aware graph tokenizer, a context-aware enhancement module, and teacher–student distillation. Across five curated multiplex benchmarks and both transductive and zero-shot settings, the method reports consistent gains in AUROC/AUPRC over single-graph, multiplex, and domain-specific baselines, with ablations attributing the largest lift to LLM embeddings.

**Strengths:**

- The authors evaluate on multiple benchmarks spanning different biological domains, suggesting improvements that generalize beyond a single dataset.

- The training/inference workflow is clearly presented and makes the role of each component (UGT, CAE, MoE, KD) easy to follow.

**Weaknesses:**

- Figure 3’s dataflow appears inconsistent with the algorithm: pooling is depicted in parallel with the graph transformer, whereas the text implies a post-transformer operation.

- Several attention mechanisms are introduced without a clear design rationale; reporting layer-wise attention weights (e.g., across multiplex layers) would help justify the need for attention aggregation.

- Some modules resemble standard variants (e.g., the “node-level context-aware attention” aligns with common attention aggregation); please delineate what is novel vs. adapted.

- All benchmarks are curated by the authors, yet basic dataset statistics and curation decisions (interaction counts by type, inclusion/exclusion filters, handling of homogeneous edges within MBNs) are missing and should be documented.

- For ChEMBL, specify the exact assay criteria/IDs used to build the antibiotic response benchmark; the current size seems very small relative to the database.

- Several baselines may be underpowered: e.g., MLP/XGB should also use the same sequence LLM embeddings to provide a fairer comparison (especially in zero-shot). Also clarify any hyperparameter search to avoid bias.

**Questions:**

- Please specify the exact model versions/checkpoints for the sequence LLMs (e.g., ESM-2 model size).

- How is edge/relational heterogeneity handled in the graph transformer—via relation-specific parameters, type encodings, or layer separation?

- How are edge perturbations performed?

- Detail the MoE design (experts, gating, placement); is it a Transformer-MoE or a lightweight MLP-expert setup?

- Describe the zero-shot split protocol.

- Where are the MolTrans results referenced as a domain baseline?

- How are bacterial entities embedded (which LLM/preprocessing)?

- Please add error bars to Figure 4 to reflect variability across interaction types.

- Clarify how graph baselines are adapted to zero-shot.

---

> ### Author Response · Authors · 2025-11-25
> **Rebuttal to Reviewer Xr4R - Part 1**
>
> Dear Reviewer,
>
> We thank you for acknowledging (i)the breadth of our evaluation across multiple biological domains, and (ii) the clarity of our training and inference workflow. Below we summarize the additional analyses we performed in response to your comments, followed by point-by-point response to the questions and weakness.
>
>
> | Concern (W/Q)             | What we did                                                            | Where in revision                          |
> |---------------------------|------------------------------------------------------------------------|--------------------------------------------|
> | W1: Fig. 3 inconsistency  | Corrected dataflow (Transformer → pooling)                            | Main text, updated Figure 3                |
> | W2: Attention rationale   | Clarified 3 attention components, reported layer-level weights         | Main rebuttal W2; Supp. Sec. E.3.7, Fig. 7 |
> | W4 / Q3: Dataset stats    | Added full stats + multiplex rate and detailed curation per dataset    | Main rebuttal W4; Supp. Sec. D, Table 8    |
> | W6: XGB/MLP Baselines    | Added XGB/MLP baselines with LLM embeddings  (T + ZS)   | Main rebuttal W5; Supplementary Section E.3.1 (Tables 10–14)|
> | Q8: Error bars (Fig. 4)   | Added error bars + dataset- and type-level ablation plots              | Main rebuttal Q8; Supp. Sec. E.3, Fig. 6   |
>
>
> **W1: Inconsistency in Figure 3**
>
> Thank you so much for pointing this out! We have **revised Figure 3 in the updated manuscript** so that the dataflow matches the algorithm precisely: graph transformer blocks first, followed by pooling.
>
> ---
>
> **W2: Rationale of Attention Mechanisms and Layer-Level Attention Weights**
>
> There are mainly three attention mechanisms:
>
> (1) **Node-level context-aware inter-layer attention**, which adaptively weights a node’s representations across multiplex layers by comparing its layer-specific embeddings. This allows the model to determine how much node $n$ in layer $p$ should attend to its counterpart in layer $q$, enabling better cross-layer alignment.
>
> (2)  **Layer-level context-aware consensus attention**, which aggregates the layer-specific node features into a shared representation $H$ and its negative counterpart $\tilde{H}$. A contrastive consensus regularizer then learns a multiplex-wide consensus embedding $Z$ by maximizing agreement with $H$ and minimizing disagreement with $\tilde{H}$. **Learning layer-level attention weights allows the model to down-weight noisy or weak layers and place more emphasis on layers that provide stronger relational signal**.
>
> (3) **Expert-gating attention in the Mixture-of-Experts (MoE) head.** For multi-label interaction prediction, we use a lightweight MoE classifier where a gating network computes a softmax over experts to produce expert-specific attention weights. This attention mechanism allows the model to combine specialized experts based on the entity-pair embedding.
>
> We hereby report the average layer-level attention weights for aggregation in (2):
>
> | Dataset       | Interaction Type                 | Weight |
> |---------------|----------------------------------|--------|
> | TRRUST        | Activation                       | 0.31   |
> | TRRUST        | Repression                       | 0.104  |
> | TRRUST        | Unknown                          | 0.585  |
> | ChEMBL        | S                                | 0.25   |
> | ChEMBL        | I                                | 0.205  |
> | ChEMBL        | R                                | 0.544  |
> | DGIdb         | Inhibitor/Blocker                | 0.366  |
> | DGIdb         | Modulator                        | 0.293  |
> | DGIdb         | Agonist                          | 0.183  |
> | DGIdb         | Activator                        | 0.366  |
> | DGIdb         | Other                            | 0.06   |
> | MetaConserve  | Campylobacterales                | 0.347  |
> | MetaConserve  | Desulfovibrionales               | 0.132  |
> | MetaConserve  | Fusobacteriales                  | 0.287  |
> | MetaConserve  | Veillonellales                   | 0.233  |
> | PINNACLE      | Enterocyte (Large Intestine)     | 0.062  |
> | PINNACLE      | Enterocyte (Small Intestine)     | 0.055  |
> | PINNACLE      | Goblet Cell (Large Intestine)    | 0.033  |
> | PINNACLE      | Goblet Cell (Small Intestine)    | 0.020  |
> | PINNACLE      | Paneth Cell (Large Intestine)    | 0.132  |
> | PINNACLE      | Paneth Cell (Small Intestine)    | 0.111  |
> | PINNACLE      | CD4⁺ Helper T Cell               | 0.068  |
> | PINNACLE      | CD4⁺ Memory T Cell               | 0.044  |
> | PINNACLE      | CD8⁺ Cytotoxic T Cell            | 0.09   |
> | PINNACLE      | Regulatory T Cell (Treg)         | 0.154  |
> | PINNACLE      | B Cell                           | 0.136  |
> | PINNACLE      | Memory B Cell                    | 0.098  |
>
> For a better illustration, please see **Supplementary Section E.3.7 and Figure 7**.

---

> ### Author Response · Authors · 2025-11-25
> **Rebuttal to Reviewer Xr4R - Part 2**
>
> **W3:  Attention Mechanisms: Novelty vs. Adapted**
>
>  We hereby summarize the three attention-related components in our framework.
>
> **(1) Node-level context-aware attention.**
>
> This is a novel application of attention for multiplex networks. The novelty is that **it aligns a node’s representations across layers using context-aware weighting**, allowing the model to integrate heterogeneous layer-specific signals.
>
> **(2) Layer-level consensus attention.**
>
> This is an adapted mechanism that applies attention at the layer level. The novelty is that **layer-specific features are selectively weighted and aggregated, allowing the model to emphasize relationship types that carry stronger or more reliable biological signals**. This adaptive weighting produces a more informative cross-layer representation, which then serves as a stronger anchor for the contrastive objective.
>
> **(3) Mixture-of-Experts (MoE) prediction head.**
>
> The MoE idea is established, but **our use of a lightweight MLP-based expert mixture with a softmax gating network is an adapted integration for multiplex multi-label prediction**, enabling specialized expert routing under heterogeneous interaction patterns.
>
> ---
>
> **W4:  Summary Statistics of Datasets and Curation Decisions**
>
> Below is the summary statistics of our five datasets. We also included this in the updated **Supplementary Section D (Table 8)**.
>
> | Dataset        | Interaction Types                                                                                                      | # Nodes | # Edges   | # Interaction Types | Multiplex Rate (%) |
> |----------------|-------------------------------------------------------------------------------------------------------------------------------|---------|-----------|----------------------|---------------------|
> | **DGIdb**      | Inhibitor (7,004); Modulator (1,439); Agonist (915); Activator (104); Other (170)                                            | 1,846   | 8,443     | 5                    | 1.34                |
> | **ChEMBL**     | S (120,265); I (33,925); R (94,392)                                                                                           | 9,368   | 248,582   | 3                    | 14.45               |
> | **PINNACLE**   | Enterocyte-LI (4,089); Enterocyte-SI (7,433); Goblet-LI (3,073); Goblet-SI (6,258); Paneth-LI (8,291); Paneth-SI (15,093); CD4-Helper (6,893); CD4-Memory (7,190); CD8-Cytotoxic (8,172); Treg (15,626); B (9,261); Memory-B (10,064) | 7,044   | 101,443   | 12                   | 23.26               |
> | **MetaConserve** | Campylobacterales (109); Desulfovibrionales (98); Fusobacteriales (94); Veillonellales (76)                                | 329     | 377       | 4                    | 27.10               |
> | **TRRUST**     | Activation (3,149); Repression (1,922); Unknown (4,325)                                                                       | 2,862   | 9,396     | 3                    | 8.90                |
>
> For curation decisions, please refer to **Supplementary Section D.1 to D.5** for details. Here is a quick summary:
>
> - **DGIdb (1,846 nodes; 8,443 edges; 5 interaction types).**
>   We removed entries lacking interaction-type annotations, merged semantically similar categories, filtered out interactions in the lowest 15% confidence quantile, and retained drugs appearing ≥7 times and genes appearing ≥4 times.
>
> - **ChEMBL (9,368 nodes; 248,582 edges; 3 interaction types).**
>   We retained only MIC-related assays convertible to unified units, kept entries with valid canonical SMILES, and applied a minimum-frequency filter keeping compounds and bacterial strains appearing **more than 11 times**. MIC values were standardized and mapped to S/I/R categories using:
>   S (≤ 4), I (4–16], R (> 16).
>
> - **PINNACLE (7,044 nodes; 101,443 edges; 12 interaction types).**
>   From the original 156 cell types, we selected 12 IBD-relevant cell types (enterocytes, goblet cells, Paneth cells, $CD4^+$ T cells, $CD8^+$ T cells, Tregs, memory T cells, B cells, and memory B cells) and constructed cell-type–specific multiplex PPIs.
>
> - **MetaConserve (329 nodes; 377 edges; 4 interaction types).**
>  We curated interactions from four IBD-associated bacterial orders (Campylobacterales, Desulfovibrionales, Fusobacteriales, Veillonellales) and retained protein–metabolite pairs with **conservation score ≥ 2**, forming four conserved interaction layers.
>
> - **TRRUST (2,862 nodes; 9,396 edges; 3 interaction types).**
>  We used the human transcription factor–target regulatory network and represented activation, repression, and unknown regulatory modes as separate interaction types.
>
> All datasets used in this study are publicly available from their original sources.

---

> ### Author Response · Authors · 2025-11-25
> **Rebuttal to Reviewer Xr4R - Part 3**
>
> **W5: ChEMBL Filtering Criteria**
>
> For ChEMBL, the benchmark is a filtered, higher-quality subset. We retained only assays with MIC-related measurements that could be converted into unified units, kept entries with valid canonical SMILES to ensure consistent compound representation, and applied a minimum-frequency filter by keeping only compounds and bacterial strains that appear more than 11 times. These steps reduce noise and sparsity, yielding a coherent dataset suitable for reliable multiplex interaction prediction rather than reflecting the full size of the ChEMBL database.
>
> We have also added the explanation to **Supplementary Section D.2** for better clarity.
>
> ---
>
> **W6: MLP/XGB with LLM Embeddings**
>
> To address this concern, we **added MLP/XGB baselines that also use the same sequence-LLM embeddings as CAZI-MBN**. These updated results are reported in **Supplementary Section E.3.1 (Tables 10–14)** for both transductive and zero-shot settings.
>
> Below is a summary table of results from XGB/MLP with sequence-LLM embedding features, as well as their comparison with CAZI-MBN.
>
> | Dataset       | Setting | Model        | AUROC           | AUPRC           | HS              | SA              |
> |---------------|---------|--------------|-----------------|-----------------|-----------------|-----------------|
> | DGIdb         | T       | XGB (LLM)    | 0.517±0.012     | 0.572±0.009     | 0.541±0.014     | 0.488±0.010     |
> | DGIdb         | T       | MLP (LLM)    | 0.511±0.011     | 0.561±0.008     | 0.536±0.010     | 0.503±0.012     |
> | DGIdb         | T       | **CAZI-MBN** | **0.715±0.007** | **0.729±0.009** | **0.687±0.011** | **0.684±0.015** |
> | DGIdb         | ZS      | XGB (LLM)    | 0.487±0.011     | 0.522±0.010     | 0.517±0.015     | 0.496±0.006     |
> | DGIdb         | ZS      | MLP (LLM)    | 0.494±0.014     | 0.525±0.006     | 0.522±0.012     | 0.492±0.007     |
> | DGIdb         | ZS      | **CAZI-MBN** | **0.671±0.008** | **0.709±0.011** | **0.688±0.009** | **0.663±0.014** |
> | ChEMBL        | T       | XGB (LLM)    | 0.648±0.014     | 0.694±0.016     | 0.735±0.010     | 0.721±0.011     |
> | ChEMBL        | T       | MLP (LLM)    | 0.624±0.008     | 0.678±0.013     | 0.718±0.011     | 0.697±0.009     |
> | ChEMBL        | T       | **CAZI-MBN** | **0.812±0.008** | **0.863±0.006** | **0.889±0.014** | **0.757±0.011** |
> | ChEMBL        | ZS      | XGB (LLM)    | 0.622±0.019     | 0.712±0.011     | 0.637±0.013     | 0.616±0.012     |
> | ChEMBL        | ZS      | MLP (LLM)    | 0.634±0.014     | 0.728±0.009     | 0.691±0.015     | 0.639±0.014     |
> | ChEMBL        | ZS      | **CAZI-MBN** | **0.791±0.018** | **0.839±0.015** | **0.857±0.011** | **0.723±0.009** |
> | PINNACLE      | T       | XGB (LLM)    | 0.742±0.010     | 0.767±0.009     | 0.744±0.007     | 0.734±0.018     |
> | PINNACLE      | T       | MLP (LLM)    | 0.733±0.011     | 0.751±0.010     | 0.699±0.010     | 0.708±0.014     |
> | PINNACLE      | T       | **CAZI-MBN** | **0.831±0.018** | **0.845±0.011** | **0.751±0.009** | **0.772±0.013** |
> | PINNACLE      | ZS      | XGB (LLM)    | 0.719±0.020     | 0.731±0.012     | 0.688±0.014     | 0.671±0.021     |
> | PINNACLE      | ZS      | MLP (LLM)    | 0.714±0.017     | 0.729±0.014     | 0.683±0.009     | 0.685±0.019     |
> | PINNACLE      | ZS      | **CAZI-MBN** | **0.812±0.013** | **0.820±0.008** | **0.748±0.010** | **0.763±0.011** |
> | MetaConserve  | T       | XGB (LLM)    | 0.662±0.009     | 0.671±0.018     | 0.658±0.020     | 0.669±0.013     |
> | MetaConserve  | T       | MLP (LLM)    | 0.694±0.012     | 0.702±0.015     | 0.707±0.016     | 0.703±0.010     |
> | MetaConserve  | T       | **CAZI-MBN** | **0.752±0.020** | **0.744±0.012** | **0.779±0.008** | **0.656±0.014** |
> | MetaConserve  | ZS      | XGB (LLM)    | 0.631±0.014     | 0.637±0.017     | 0.635±0.009     | 0.633±0.012     |
> | MetaConserve  | ZS      | MLP (LLM)    | 0.659±0.018     | 0.662±0.016     | 0.664±0.012     | 0.672±0.014     |
> | MetaConserve  | ZS      | **CAZI-MBN** | **0.738±0.015** | **0.722±0.010** | **0.749±0.020** | **0.653±0.009** |
> | TRRUST        | T       | XGB (LLM)    | 0.854±0.019     | 0.863±0.027     | 0.852±0.014     | 0.867±0.018     |
> | TRRUST        | T       | MLP (LLM)    | 0.861±0.022     | 0.868±0.031     | 0.856±0.017     | 0.859±0.021     |
> | TRRUST        | T       | **CAZI-MBN** | **0.905±0.013** | **0.872±0.008** | **0.791±0.023** | **0.784±0.015** |
> | TRRUST        | ZS      | XGB (LLM)    | 0.828±0.143     | 0.836±0.198     | 0.842±0.117     | 0.831±0.022     |
> | TRRUST        | ZS      | MLP (LLM)    | 0.845±0.009     | 0.852±0.017     | 0.837±0.021     | 0.851±0.015     |
> | TRRUST        | ZS      | **CAZI-MBN** | **0.899±0.017** | **0.869±0.013** | **0.775±0.019** | **0.764±0.007** |

---

> ### Author Response · Authors · 2025-11-25
> **Rebuttal to Reviewer Xr4R - Part 4**
>
> As expected, using LLM embeddings boosts the performance of XGB and MLP, which is consistent with our ablation findings showing that sequence LLMs provide strong domain-informed representations. Even with this improvement, however, **these baselines still fall short of CAZI-MBN**. This shows that sequence features are helpful but sometimes not sufficient on their own, and that additional modeling choices, such as incorporating multiplex structure, are needed to reach the performance we observe with CAZI-MBN.
>
> As clarified in **Supplementary Section E.2**, We performed grid-search hyperparameter search for the models.
>
> ---
> **Q1:  LLM Model Checkpoints**
>
> | **Model**      | **Checkpoint**                | **Available At** |
> |----------------|-------------------------------|------------------|
> | DNABERT-2      | 117M                           | https://huggingface.co/zhihan1996/DNABERT-2-117M |
> | ESM-2          | 150M                           | https://huggingface.co/facebook/esm2_t30_150M_UR50D |
> | ChemBERTa      | 77M                            | https://huggingface.co/DeepChem/ChemBERTa-77M-MLM |
>
> We have also added these details to **Supplementary Section A**.
>
> ---
> **Q2: Handling of Heterogeneous Edges**
>
> Multiplex networks contain homogeneous edges within each layer but heterogeneous semantics across layers. In CAZI-MBN, heterogeneity is handled by assigning each interaction type to a separate Graph Transformer block with distinct parameters. This preserves relation-specific structure and prevents mixing heterogeneous edge semantics during message passing.
>
> ---
> **Q3: Edge Perturbation**
>
>  For each layer, we create a perturbed graph by randomly sampling non-edges from the same node set. We sample pairs $(u, v)$ uniformly, remove duplicates and existing edges, exclude self-loops, and match the negative sample size to the original number of edges. This produces a structurally corrupted counterpart used for contrastive learning.
>
> ---
> **Q4: MoE Architecture**
>
> Our MoE is a lightweight MLP-expert module used in the final prediction head, not a Transformer-MoE. For each entity pair, we concatenate their embeddings and pass the vector through a set of small two-layer MLP experts. A simple gating network (a linear layer followed by softmax) produces mixture weights over these experts, and the final output is their weighted sum. We avoid Transformer-level MoE routing, as it would add substantial computational and memory overhead, whereas this lightweight design keeps the backbone unchanged and introduces only minimal additional cost.
>
> ---
> **Q5: Zero-Shot Split Protocol**
>
> We described our zero-shot split protocol in **Supplementary Section E.2**.  In the zero-shot setting, we use a strict entity-disjoint split. We randomly divide the nodes into training, validation, and test groups in proportions of 70%, 15%, and 15%. All edges connected to validation or test nodes are removed from the training graph, so these nodes never appear as endpoints or neighbors during message passing. As a result, the model cannot use any structural information about them during training, and all predictions must be made only from their sequence embeddings. This setup avoids any information leakage and reflects realistic situations where new drugs, genes, or species enter the system with no known interactions.
>
> ---
> **Q6: MolTrans**
>
> MolTrans is a domain-specific for MetaConserve. Its results are included in **Supplementary Section E.3.1 Table 13**.
>
> ---
> **Q7: LLM Embeddings for Bacteria Entities**
> For bacteria, their genomes are segmented into fixed-length windows (length=1024) and fed into DNABERT-2. We average the embedding outputs across windows to obtain a compact genome-level representation. We have added this preprocessing step to **Supplementary Section E.2** for better clarity.
>
> ---
> **Q8: Error Bars in Ablation Figures**
>
> Figure 4 reports the average performance drop across datasets in our ablation study. To better convey variability, we have **added error bars that capture the variation in metric drops across different datasets**. In response to the suggestion, we added a plot that shows the module-wise performance drops across all datasets with error bars illustrating the variance across interaction types in **Figure 6 in Supplementary Section E.3.**
>
> ---
> **Q9: Zero-Shot Baseline Adaption**
>
> As documented in **Supplementary Section E.2:**  In zero-shot setting, where traditional graph-based models cannot inherently predict interactions involving entities with no observed training interactions, we adopt knowledge-distilled variants of these models. Specifically, the original graph-based models serve as teacher models, while a two-layer MLP acts as the student model.
>
> ---
> We hope that this addresses the concerns and help improve clarity. We thank the reviewer again for your thorough review and suggestion.
>
> Authors

---

> > ### Author Response · Authors · 2025-11-28
> > **Hope You Can Take a Look at Our Responses**
> >
> > Dear Reviewer,
> > We hope this message finds you well. As the discussion period is approaching its end, we wanted to kindly draw your attention to our responses to your valuable comments. We hope they address your concerns, and we would greatly appreciate any additional thoughts or clarifications you may have.
> >
> > Your feedback is invaluable, and we are eager to refine our work based on your insights. If there are any remaining points you would like us to consider, please feel free to let us know.
> >
> > Thank you very much for your time and effort in reviewing our paper.
> >
> > The authors

---

### Official Review · Reviewer_7pwj · 2025-10-31

**Soundness:** 2
**Presentation:** 2
**Contribution:** 3
**Rating:** 6
**Confidence:** 3

**Summary:**

This paper proposes CAZI-MBN, a topology-aware teacher–student framework for interaction prediction on multiplex biological networks (MBNs), targeting in particular the zero-shot case where new entities have no observed neighborhood (e.g., new drugs, uncharacterized genes). The teacher model combines (i) domain LLM–based sequence embeddings (ChemBERTa-2, DNABERT-2, ESM-2), (ii) topology embeddings from a Unified Graph Tokenizer over the supra-adjacency, and (iii) a Context-Aware Enhancement (CAE) module with graph transformers and contrastive/consensus objectives. A Mixture-of-Experts head is used for multi-label interaction prediction across relation types. A topology-agnostic student is then distilled to match the teacher’s latent representations and task outputs, so that it can make zero-shot predictions from sequence only. The authors also curate five multiplex benchmarks (DGIdb, ChEMBL, PINNACLE, MetaConserve, TRRUST) to standardize evaluation in this setting.

**Strengths:**

1. The teacher-student distillation idea is well matched to the zero-shot MBN problem: topology is used only when available, but its signal is transferred to a sequence-only student so that unseen entities can still be scored.

2. The architecture is thoughtfully composed (LLM sequences + UGT topology + CAE alignment + MoE for multi-label), and the ablations suggest that CAE and LLM features contribute the most.

3. The IBD case study shows that the method can surface biologically plausible candidates, not just improve AUROC.

**Weaknesses:**

1. The full teacher pipeline is heavy: it requires multiple domain LLMs, SVD/factorization on a supra-adjacency matrix for UGT, and graph transformers within CAE. The paper does not report runtime / GPU hours / UGT rank on the largest MBN, so it is hard to assess scalability to very large networks.

2. On smaller / skewed datasets such as TRRUST, the model improves AUROC but underperforms some baselines on Hamming Score / Subset Accuracy, suggesting that label imbalance, MoE calibration, or the thresholding strategy may affect multi-label metrics. A short analysis would make the empirical section tighter.

3. Several baselines are adapted to the multiplex setting by training a separate classifier per interaction type. This is a reasonable but relatively weak adaptation. Adding at least one heterogeneous GNN (e.g., an R-GCN-style model or a multi-head GAT over relation types) would strengthen the comparison.

**Questions:**

1. Please report GPU hours, parameter counts (teacher vs. student), UGT rank, and runtime per epoch on the largest dataset.

2. Can you add or report a native multiplex/heterogeneous GNN baseline to show that CAZI-MBN is competitive even against relation-aware models?

3. For datasets where CAZI-MBN lags on HS/SA, how were decision thresholds chosen (global vs. per-label vs. per-dataset)? Would per-label calibration close the gap?

---

> ### Author Response · Authors · 2025-11-25
> **Rebuttal to Reviewer 7pwj - Part 1**
>
> Dear Reviewer,
>
> Thank you so much for your constructive review. We appreciate your recognition of the key strengths of our work: (i) the teacher–student distillation design, where topology is used only when available but its signal is transferred to a sequence-only student for zero-shot prediction; (ii) the coherent architecture combining LLM sequences, UGT topology, CAE alignment, and MoE; and (iii) the IBD case study, which shows that the approach can surface biologically sensible interaction candidates.
>
> To help clarify the scope of our experimental effort, we begin with a short summary of the additional analyses performed in response to your concerns:
>
> - **(W1/Q1)** Added detailed GPU-hours, parameter counts, UGT orders, and per-epoch runtimes for all five datasets (Supplementary Section E.3.5, Table 19).
> - **(W3/Q2)** Added two relation-aware GNN baselines (R-GCN and R-GAT) alongside existing multiplex baselines; evaluated both in transductive and zero-shot settings (Supplementary Tables 10–14).
> - **(W2/Q3)** Conducted a per-label calibration analysis on TRRUST for both CAZI-MBN and all baselines; observed clear improvements in HS and SA (Supplementary Section E.3.6, Table 20).
>
> Below, we respond to each of your comments in detail.
>
> **1. W1/Q1: GPU hours, parameter counts, UGT rank, and runtime per epoch**
>
> The table below summarizes the dataset sizes, parameter counts, average GPU time (over 3 runs), and average runtime per epoch. Because we use early stopping, the total number of epochs varies across runs.
> | Dataset       | # Nodes | # Interaction Types | UGT Order | Teacher Total GPU Hours | Teacher Time/Epoch | Student Total GPU Hour | Student Time/Epoch | Teacher Parameters | Student Parameters |
> |---------------|---------|--------------|-----------|-------------------------|---------------------|-------------------------|---------------------|----------------------|----------------------|
> | MetaConserve  | 329     | 4            | 5         | 0.463 h   | 0.574 s             | 0.00109 h | 0.005 s             | 2016.820 M           | 1.189 M              |
> | DGIdb         | 1,846   | 5            | 5         | 3.191 h    | 4.332 s             | 0.0567 h | 0.129 s             | 3430.584 M           | 1.734 M              |
> | TRRUST        | 2,862   | 3            | 5         | 2.137 h   | 3.903 s             | 0.0098 h      | 0.078 s             | 2062.506 M           | 1.041 M              |
> | PINNACLE      | 7,044   | 12           | 5         | 31.34 h       | 115.307 s               | 0.4275 h | 3.802 s               | 6045.610 M           | 3.568 M              |
> | ChEMBL        | 9,368   | 3            | 5         | 8.456 h | 43 .201s                | 0.114 h | 1.3 11s               | 2068.336 M           | 1.041 M              |
>
> The table shows that the student model is much smaller and trains far faster than the teacher. In practice, the student uses only a small portion of the total GPU time because it has far fewer parameters and avoids the heavier graph operations. One thing to note is that **teacher runtime scales more with the number of interaction types than with node count**, since the model builds a separate graph transformer for each type. This is why PINNACLE takes longer to train than ChEMBL despite having fewer nodes. This analysis has also been added to the updated manuscript as **Table 19 in Supplementary Section E.3.5.**
>
> ---
>
>
> **2. W3/Q2: Multiplex/heterogeneous GNN Baselines**
>
> Our evaluation **already includes several established multiplex models**, including MultiplexSAGE, DMGI, and HDMI, which directly operate on multiplex graph structure. These models were originally designed for general multiplex networks and are typically evaluated on node-level tasks such as node classification or community detection, rather than on interaction prediction. **To further strengthen our comparison and respond to the reviewer’s suggestion, we additionally incorporate two relation-aware GNN baselines: (1)R-GCN [1]; (2)R-GAT[2].**
>
> For both models, each multiplex layer is treated as a distinct relation type. The R-GCN-style baseline performs relation-specific message passing by applying a separate transformation to each relation type and aggregating the resulting messages into unified node embeddings. The R-GAT-style baseline also computes relation-specific attention coefficients on these relation-labeled edges and combines the weighted messages to update node embeddings. For zero-shot generalization, we follow the same setup as with our other graph-based baselines by using these GNNs as teachers and a simple MLP classifier chain as the student.

---

> ### Author Response · Authors · 2025-11-25
> **Rebuttal to Reviewer 7pwj - Part 2**
>
> Please see below the table for the updated baselines and the comparison to our CAZI-MBN model. The results are also added to **Supplementary Section E.3.1 Table 10 to 14**.
>
>
> | Dataset       | Setting | Model       | AUROC          | AUPRC          | HS             | SA             |
> |---------------|---------|-------------|-----------------|-----------------|----------------|----------------|
> | DGIdb         | T       | R-GCN       | 0.536±0.018     | 0.541±0.009     | 0.530±0.014    | 0.508±0.011    |
> | DGIdb         | T       | R-GAT       | 0.544±0.007     | 0.549±0.012     | 0.521±0.017    | 0.514±0.021    |
> | DGIdb         | T       | **CAZI-MBN** | **0.715±0.007** | **0.729±0.009** | **0.687±0.011**| **0.684±0.015**|
> | DGIdb         | ZS      | R-GCN       | 0.510±0.015     | 0.514±0.008     | 0.525±0.028    | 0.500±0.013    |
> | DGIdb         | ZS      | R-GAT       | 0.512±0.012     | 0.518±0.014     | 0.519±0.015    | 0.507±0.020    |
> | DGIdb         | ZS      | **CAZI-MBN**    | **0.671±0.008** | **0.709±0.011** | **0.688±0.009**| **0.663±0.014**|
> | ChEMBL        | T       | R-GCN       | 0.647±0.016     | 0.725±0.011     | 0.760±0.031    | 0.719±0.023    |
> | ChEMBL        | T       | R-GAT       | 0.655±0.013     | 0.742±0.009     | 0.771±0.025    | 0.722±0.010    |
> | ChEMBL        | T       | **CAZI-MBN**   | **0.812±0.008** | **0.863±0.006** | **0.889±0.014**| **0.757±0.011**|
> | ChEMBL        | ZS      | R-GCN       | 0.630±0.013     | 0.719±0.009     | 0.714±0.018    | 0.687±0.012    |
> | ChEMBL        | ZS      | R-GAT       | 0.643±0.026     | 0.731±0.012     | 0.720±0.007    | 0.708±0.017    |
> | ChEMBL        | ZS      | **CAZI-MBN**    | **0.791±0.018** | **0.839±0.015** | **0.857±0.011**| **0.723±0.009**|
> | PINNACLE      | T       | R-GCN       | 0.761±0.017     | 0.786±0.012     | 0.754±0.024    | 0.753±0.019    |
> | PINNACLE      | T       | R-GAT       | 0.759±0.009     | 0.783±0.013     | 0.760±0.011    | 0.743±0.015    |
> | PINNACLE      | T       | **CAZI-MBN**    | **0.831±0.018** | **0.845±0.011** | **0.751±0.009**| **0.772±0.013**|
> | PINNACLE      | ZS      | R-GCN       | 0.733±0.032     | 0.765±0.019     | 0.742±0.012    | 0.733±0.017    |
> | PINNACLE      | ZS      | R-GAT       | 0.726±0.023     | 0.758±0.020     | 0.734±0.025    | 0.735±0.016    |
> | PINNACLE      | ZS      |**CAZI-MBN**    | **0.812±0.013** | **0.820±0.008** | **0.748±0.010**| **0.763±0.011**|
> | MetaConserve  | T       | R-GCN       | 0.712±0.017     | 0.718±0.010     | 0.698±0.013    | 0.703±0.009    |
> | MetaConserve  | T       | R-GAT       | 0.720±0.028     | 0.723±0.017     | 0.714±0.013    | 0.710±0.014    |
> | MetaConserve  | T       | **CAZI-MBN**    | **0.752±0.020** | **0.744±0.012** | **0.779±0.008**| **0.656±0.014**|
> | MetaConserve  | ZS      | R-GCN       | 0.699±0.021     | 0.702±0.012     | 0.683±0.016    | 0.685±0.010    |
> | MetaConserve  | ZS      | R-GAT       | 0.703±0.019     | 0.705±0.022     | 0.691±0.014    | 0.699±0.018    |
> | MetaConserve  | ZS      | **CAZI-MBN**    | **0.738±0.015** | **0.722±0.010** | **0.749±0.020**| **0.653±0.009**|
> | TRRUST        | T       | R-GCN       | 0.865±0.011     | 0.867±0.013     | 0.861±0.020    | 0.870±0.015    |
> | TRRUST        | T       | R-GAT       | 0.868±0.019     | 0.871±0.014     | 0.866±0.018    | 0.875±0.023    |
> | TRRUST        | T       | **CAZI-MBN**    | **0.905±0.013** | **0.872±0.008** | **0.791±0.023**| **0.784±0.015**|
> | TRRUST        | ZS      | R-GCN       | 0.854±0.013     | 0.856±0.009     | 0.850±0.017    | 0.861±0.012    |
> | TRRUST        | ZS      | R-GAT       | 0.859±0.015     | 0.863±0.012     | 0.859±0.019    | 0.863±0.017    |
> | TRRUST        | ZS      | **CAZI-MBN**   | **0.899±0.017** | **0.869±0.013** | **0.775±0.019**| **0.764±0.007**|
>
> Across all datasets, **the performance of the relation-aware GNNs (R-GCN and R-GAT) generally falls between the strongest multiplex-graph baselines and the best single-graph baselines**. Relation-aware GNNs do benefit from using relation-specific message passing, but they do not capture the full multiplex structure as most multiplex-graph-based models. **Overall, our proposed model has better performance than these relation-aware GNNs for all the tested datasets.**

---

> ### Author Response · Authors · 2025-11-25
> **Rebuttal to Reviewer 7pwj - Part 3**
>
> **3. W2/Q3: Per-Label Calibration**
>
> Thank you for raising this point. In our main results, decision thresholds were chosen using a global value of 0.5 for all interaction types. We agree that per-label calibration is a good strategy for datasets where the model lags on Hamming Score (HS) and Subset Accuracy (SA). In response, we conducted a **post-hoc per-label calibration experiment** on the dataset where CAZI-MBN shows relatively lower HS and SA: **TRRUST**. For each interaction type, we swept thresholds from 0.2 to 0.8 in increments of 0.1 and selected the value that maximized validation accuracy for each interaction type. The same post-hoc per-label calibration strategy is also applied on the baselines to ensure fair comparison.
>
> After applying per-label calibration, we see that most models gain in HS and SA, and CAZI-MBN also shows quite noticeable improvement in HS/SA on TTRUST. This shows that using label-specific thresholds can help reduce the gap in cases where CAZI-MBN scores lower on these metrics, and it offers a practical way to make the predictions sharper. Below is a brief summary table showing the changes in CAZI-MBN on TRRUST.
>
> | Setting | Model                              | HS                 | SA                 |
> |---------|-------------------------------------|--------------------|--------------------|
> | **T**   | **CAZI-MBN (Per-Label Calibration)** | 0.886±0.015    | 0.891±0.014 |
> | **T**   | **CAZI-MBN (Unified Threshold)**     | 0.791±0.023    | 0.784±0.015  |
> | **ZS**  | **CAZI-MBN (Per-Label Calibration)** | 0.882±0.013   |0.884±0.020|
> | **ZS**  | **CAZI-MBN (Unified Threshold)**     | 0.775±0.019 | 0.764±0.007 |
>
> ---
>
> Again, we really appreciate the reviewer for pointing out this very helpful direction.
> We have updated the manuscript accordingly. The full results, including per-label calibration for the baselines, are now provided in **Supplementary Section E.3.6**. We also added a short discussion in the main paper that points readers to this analysis.
>
> We thank you again for the solid suggestions and your review and we hope that our response address your concerns
>
> Authors
>
> ---
> References:
>
> [1] Schlichtkrull, Michael, et al. "Modeling relational data with graph convolutional networks." European semantic web conference. Cham: Springer International Publishing, 2018.
>
> [2] Chen, Meiqi, et al. "r-gat: Relational graph attention network for multi-relational graphs." arXiv preprint arXiv:2109.05922 (2021).

---

> > ### Author Response · Authors · 2025-11-28
> > **Hope You Can Take a Look at Our Responses**
> >
> > Dear Reviewer,
> > We hope this message finds you well. As the discussion period is approaching its end, we wanted to kindly draw your attention to our responses to your valuable comments. We hope they address your concerns, and we would greatly appreciate any additional thoughts or clarifications you may have.
> >
> > Your feedback is invaluable, and we are eager to refine our work based on your insights. If there are any remaining points you would like us to consider, please feel free to let us know.
> >
> > Thank you very much for your time and effort in reviewing our paper.
> >
> > The authors

---

### Author Response · Authors · 2025-12-03
**Final Remarks to the Area Chair**

Dear Area Chair,

Thank you for your time reviewing our submission. We understand that this year’s process has been organized differently, and we appreciate your effort. We remained fully engaged throughout the rebuttal period and responded in detail to every reviewer’s questions. For clarity in your evaluation, we summarize the core motivation of the paper, the reviewers’ assessments, and the additional work completed during rebuttal.

---

### (1) Core Motivation and Novelties of Our Paper

Our work introduces **CAZI-MBN**, the first framework explicitly designed for **zero-shot interaction prediction in multiplex biological networks**. Multiplex networks contain multiple interaction types between the same set of entities (genes, compounds, proteins, metabolites), yet existing methods have challenges in handling **multiplex networks in biological contexts** since they:

- ignore multiplexity or flatten it,
- fail to integrate sequence embeddings with topology, and
- lack an explicit strategy for generalizing to new entities with no observed neighborhood information.

CAZI-MBN addresses these gaps through four innovative components:

 **(a) Multiplex-aware, high-order topology representation**

We introduce a **Unified Graph Tokenizer (UGT)** that encodes multiplexity and higher-order topology via smoothed supra-adjacency matrices.

**(b) Multiplex-aware representation enhancement via CAE**

The **Context-Aware Enhancement (CAE)** module models cross-layer dependencies using node-level and layer-level context-aware attention and a contrastive consensus objective. This aligns representations across interaction types, suppresses noisy relational signals, and produces a coherent multiplex-wide embedding.

**(c) Mixture-of-Experts (MoE) interaction head**

A lightweight expert-gating design models heterogeneous label dependencies across interaction types.

**(d) Topology-to-sequence knowledge distillation for zero-shot inference**

A topology-aware teacher distills to a topology-agnostic student, enabling prediction for unseen genes, proteins, and compounds using only sequence embeddings.

These components together support strong transductive and zero-shot performance over all evaluated baselines.

---

### (2) Reviewer Ratings and Assessments

**Table A. Reviewer Summary**

| Reviewer | Original Rating | Comments |
|-|-|-|
| **7pwj** | 6 | Highlighted strong teacher–student design, effective multiplex modeling, meaningful case study, and LLM/CAE contributions; suggested R-GCN baselines, calibration tests, and detailed runtime and parameter count reporting. |
| **Xr4R** | 4 | Acknowledged clear workflow, multi-domain evaluation, and interpretable components; suggested improved figures, attention justification, expanded dataset statistics, error bars, and stronger XGB/MLP baselines. |
| **EFYv** | 6 | Supported novelty of CAZI-MBN and consistent performance gains; suggested clearer negative sampling, zero-shot behavior under varying graph density, and expanded dataset descriptions. |


Overall, reviewers expressed strong support for our algorithm and approach. Their comments focused mainly on requests for clearer explanation and some additional analysis. The ratings were positive, with an overall average of **5.33**. All points have been addressed through detailed explanations and substantial new experiments incorporated into the revised manuscript.

---

### (3) Summary of Rebuttal

We conducted extensive additional analyses to directly address each reviewer concern. The table below summarizes newly added work, the issues addressed, and their corresponding locations in the revised submission.

 **Table B. Summary of New Analysis and Issues Addressed**

| New Analysis Added | Questions Addressed |
|-|-|
| Full GPU hours, epoch time, UGT order, and parameter counts for all datasets | 7pwj – W1/Q1 |
| Two relation-aware baselines (R-GCN, R-GAT), evaluated in both transductive and zero-shot | 7pwj – W3/Q2 |
| Post-hoc per-label calibration on TRRUST | 7pwj – W2/Q3 |
| Interaction-type attention weight reporting | Xr4R – W2 |
| Expanded dataset curation: multiplex rate, inclusion rules, detailed statistics | Xr4R – W4; EFYv – Q2/Q3 |
| XGB/MLP baselines upgraded with LLM embeddings | Xr4R – W6 |
| Error bars and dataset/type-level ablation plots | Xr4R – Q8 |

All results have been integrated into the revised manuscript, strengthening fairness, clarity, and transparency of the evaluation.

---

We appreciate the reviewers’ feedback and have put considerable effort into addressing all points through new baselines and analysis, calibration studies, expanded ablations, clarified architectural components, and more detailed reporting. We hope this summary is useful for your evaluation.

Thank you again for your time and consideration. We genuinely hope you will consider our submission, and we are happy to provide any further clarification if needed.

Sincerely,
Authors

---

### Meta-Review · Area_Chair_KFZR · 2026-01-07

**Summary:**

The paper proposes CAZI-MBN, a novel topology-aware framework designed for zero-shot interaction prediction in Multiplex Biological Networks (MBNs). It addresses the limitations of existing methods in handling multiplexity and generalizing to entities with no prior neighborhood information by integrating domain-specific foundation models, a Unified Graph Tokenizer (UGT), and a teacher-student distillation strategy. The authors have significantly strengthened the paper through the rebuttal process by adding extensive new experimental results and detailed documentation of the computational complexity and dataset statistics.

**Reviewer Concerns:**

Reviewer 7pwj praised the teacher-student design and the architecture's composition but raised concerns regarding scalability (runtime/GPU hours) and the strength of the baselines used. In the rebuttal, the authors reported full GPU hours, parameter counts, and runtime per epoch for all five datasets, clarifying the efficiency of the student model.

Reviewer Xr4R noted clear workflows but criticized the lack of design rationale for attention mechanisms, missing dataset statistics, and potentially underpowered baselines. The authors addressed these concerns by adding comparisons against relation-aware GNN baselines (R-GCN and R-GAT), demonstrating that CAZI-MBN consistently outperforms these models in both transductive and zero-shot settings. They also provided a detailed rationale for the three attention mechanisms used and reported layer-level attention weights to justify the multi-layered approach.

Reviewer EFYv supported the novelty and performance gains but requested clarification on negative sampling and the impact of graph density on zero-shot performance. The authors documented full dataset statistics, curation decisions, and implemented XGB/MLP baselines using the same LLM embeddings for a fairer comparison.

**Reviewer Scores:**

The authors have significantly strengthened the paper through the rebuttal process by adding extensive new experimental results and detailed documentation of the computational complexity and dataset statistics. The reviewers’ scores are all predicted to improve to at least positive.

---

### Decision · Program_Chairs · 2026-01-26

Accept (Poster)